# Long-Form Answers to Visual Questions
# from Blind and Low Vision People

**Mina Huh**[♡], **Fangyuan Xu**[♡], **Yi-Hao Peng**[♠], **Chongyan Chen**[♡], **Hansika Murugu**[♣],
**Danna Gurari**[◇], **Eunsol Choi**[♡], **Amy Pavel**[♡]

[♡]The University of Texas at Austin, [♠]Carnegie Mellon University,
[♣] Hong Kong University of Science and Technology [◇]University of Colorado Boulder

minahuh@cs.utexas.edu

## Abstract

Vision language models can now generate long-form answers to questions about images – long-form visual question answers (LFVQA). We contribute *VizWiz-LF*[1], a dataset of long-form answers to visual questions posed by blind and low vision (BLV) users. *VizWiz-LF* contains 4.2k long-form answers to 600 visual questions, collected from human expert describers and six VQA models. We develop and annotate functional roles of sentences of LFVQA and demonstrate that long-form answers contain information beyond the question answer such as explanations and suggestions. We further conduct automatic and human evaluations with BLV and sighted people to evaluate long-form answers. BLV people perceive both human-written and generated long-form answers to be plausible, but generated answers often hallucinate incorrect visual details, especially for unanswerable visual questions (*e.g.*, blurry or irrelevant images). To reduce hallucinations, we evaluate the ability of VQA models to abstain from answering unanswerable questions across multiple prompting strategies.

## 1 Introduction

Traditional visual question answering (VQA) models respond to visual questions about images with short answers. This is because they were designed for mainstream dataset challenges for which answers typically are brief, containing one or two words (Antol et al., 2015; Gurari et al., 2018; Goyal et al., 2017). The rise of large vision language models (VLMs) has introduced a new class of VQA models that can generate *long-form answers* (Alayrac et al., 2022; Team et al., 2023; Driess et al., 2023; Chen et al., 2023c; Achiam et al., 2023). The progress in VLMs meets a long-standing need from VQA solutions (Chen et al., 2023b). For instance, some questions necessitate a long-form answer, such as *"How do I make this boxed cake?"* Also, some users find answers with nuanced information to be helpful, such as answers with explanations and supplementary details (Xu et al., 2022; Naik et al., 2023). While long-form visual question answering (LFVQA) has enormous potential, we have limited knowledge about the content and quality of long-form answers.

Our work aims to understand the content and quality of long-form answers to visual questions asked by blind and low vision (BLV) people. LFVQA can be particularly impactful for BLV people who take pictures and ask questions to overcome real visual accessibility barriers. In fact, BLV people are already active consumers of long-form VQA applications like Be My AI (BeMyEyes, 2020) powered by GPT-4V. Authentic VQA involves unique challenges not present in artificial settings, such as conversational questions and low-quality images due to blur and framing errors. We seek to understand the potential and limitations of LFVQA for BLV users given its societal impact.

---

[1]Our data and code are available at https://github.com/utcshci/lfvqa.

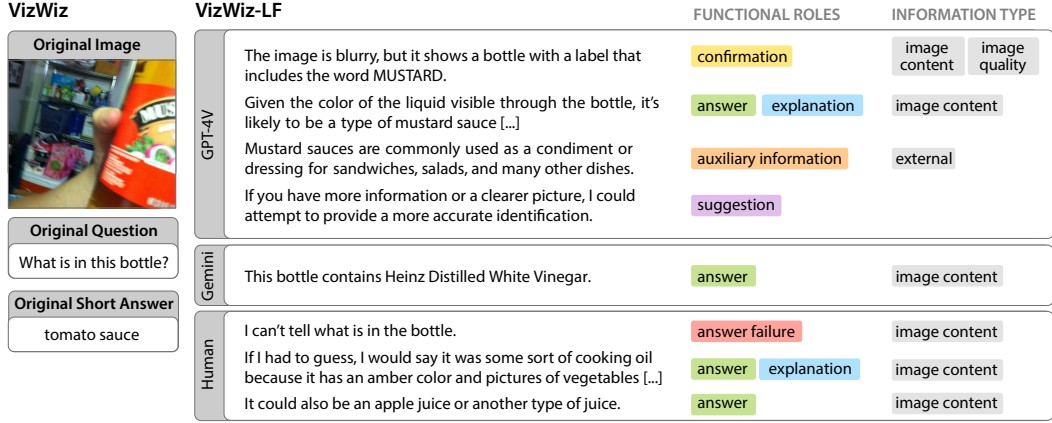

Figure 1: We collect long-form answers for image-question pairs in the VizWiz dataset (Gurari et al., 2018). For each visual question, we collect an answer from human expert describers and 6 VLMs (GPT-4V, Gemini, LLaVA, InstructBLIP, QWEN, BLIP-2). To understand the content and structure of long-form visual question answers, we create a taxonomy of functional roles and annotate answer functional roles and information types at a sentence level.

We introduce *Vizwiz-LF*, a dataset of 4.2k long-form answers to visual questions from BLV people seeking visual assistance (Bigham et al., 2010; Gurari et al., 2018). We collect and evaluate long-form answers from human experts and six state-of-the-art VQA models (GPT-4V (Achiam et al., 2023), Gemini (Team et al., 2023), LLaVA (Liu et al., 2024), InstructBLIP (Dai et al., 2024), Qwen-VL-Chat (Bai et al., 2023), and BLIP-2 (Li et al., 2023a)).

To understand the content of LFVQA, we design and annotate the communicative roles (*e.g.,* answer, explanation, suggestion) and information sources (*e.g.,* image content, image quality, or external information) of long-form answer sentences in our dataset. We create a classifier for the functional roles and information sources in LFVQA that performs on par with humans then annotate our dataset with this classifier (example in Figure 1). While the majority of answers from 5 VLMs (Gemini, LLaVA, InstructBLIP, QWEN, BLIP-2) contained only two functional roles (confirmation, answer), human expert describers and GPT-4V answers frequently included many functional roles (*e.g.,* explanation, suggestion, auxiliary information). Most long-form answers described image content, but human experts and GPT-4V also frequently described image quality.

To assess the performance of VLMs in LFVQA, we conduct an automatic evaluation of long-form answers using reference-based metrics (ROUGE (Lin, 2004), METEOR (Elliott & Keller, 2013), BERTScore (Zhang et al., 2019) and LAVE (Mañas et al., 2023)) with short-form reference answers from VizWiz and long-form reference answers from VizWiz-LF. Reference-based VQA evaluations typically use short reference answers and thus penalize long answers for including extra information (Krishna et al., 2021; Xu et al., 2023). We show that extracting answer sentences from long answers using our classifier can mitigate this.

To understand how humans evaluate long-form answers, we conduct an evaluation study with both sighted and BLV people. We examined the effect of visual priming bias by evaluating with four different conditions of {BLV, sighted}×{with image, without image}.[2] Our results also reveal that sighted people's evaluation without an image is not a strong proxy for BLV preferences. For instance, BLV evaluators tend to perceive incorrect answers as more plausible and rate extraneous details as less relevant than sighted evaluators.

As VLMs often hallucinate with answer sentences for unanswerable visual questions, we assess six VLMs' ability to abstain from providing an incorrect answer. To determine whether a model abstains, we use our functional role classifier to check for the *Answer*

---

[2]For BLV users, we simulate "with image condition" by providing them a description of the image from the original VizWiz dataset.

*Failure* role in the long-form answer, which expresses a model's inability to answer the question. Only GPT-4 achieved high recall for abstention (0.82) followed by QWEN (0.42) with default settings. We further experiment with multiple abstention approaches and reveal that prompting VLMs with abstain instructions can reduce hallucinations.

While this work addresses visual questions posed by BLV people, our findings can inspire broader research in VQA. First, our introduction of long-form answers to supplement the short answers in the original VizWiz dataset results in the first dataset with both short and long answers. Potential downstream benefits include enabling the transfer from short-answer VQA tasks, at which current models already excel, to long-answer VQA tasks. Second, our proposed LFVQA functional roles can support improvement and evaluation of LFVQA for our use case and beyond. Finally, our human evaluation highlights the importance of evaluating LFVQA beyond factual accuracy (the focus of short-form VQA) to further consider end users' experience with metrics such as relevance and plausibility.

## 2   Related Work

**VQA Datasets**   Most existing VQA datasets have brief answers consisting of one to two words collected from crowd workers (Antol et al., 2015; Zhu et al., 2016; Gurari et al., 2018; Wang et al., 2017; Krishna et al., 2017; Wu et al., 2017). This includes the widely-studied dataset with questions asked by BLV individuals, VizWiz-VQA (Gurari et al., 2018), even though BLV people often explicitly ask for detailed answers (*e.g., "Yes this is a Google Street View image, I need a detail, a detail of the description, as much as possible."* – user question from VizWiz dataset). To explore the potential of LFVQA in accessibility, we gather and analyze a dataset of detailed, sentence-to-paragraph-long answers to VizWiz questions collected from expert describers and 6 modern VLMs.

A few VQA datasets also provide longer responses: ContextVQA (avg. 11 words) (Naik et al., 2023), ScienceQA (Saikh et al., 2022) (avg. 4 words), and VQAOnline (Chen et al., 2023b) (avg. 173 words). While the answers in these datasets are collected online or from crowd workers, we collect high-quality answers from experts proficient in image description tasks. We also address questions posed by BLV individuals, which cover a different distribution of images (*e.g.,* quality) and questions (*e.g.,* describing physical objects).

**Long-form Question Answering (LFQA)**   LFQA systems (Fan et al., 2019; Nakano et al., 2021) generate comprehensive, paragraph-level answers to text-only questions. Prior work (Xu et al., 2022) analyzes the content types of human-written long-form answers from online community forums (Fan et al., 2019; Nakano et al., 2021), finding that they convey rich information beyond simply answering the questions. They propose six functional roles for answer sentences, such as "example" and "auxiliary information". As we develop functional roles for LFVQA generated by both humans and models, we reveal new roles (*e.g.,* confirmation of photos, answer failure) and different role distributions across experts and models. We also annotate information sources of LFVQA answers to assess how models reference images (*e.g.,* image content, image quality).

**Evaluating VQA Models**   Most traditional automatic VQA evaluation metrics are reference-based (Papineni et al., 2002; Lin, 2004; Elliott & Keller, 2013; Zhang et al., 2019). Yet, Krishna et al. (2021) and  Xu et al. (2023) highlighted the difficulty of evaluating long-form answers using these metrics designed for short-form answers, as they poorly correlate with human judgments. Recent LLM-based metrics (Chan et al., 2023; Liu & Chen, 2023; Kim et al., 2023; Huang et al., 2024) align better with human judgments, yet have not been explored in LFVQA. Recently, Jing et al. (2023) introduced a reference-free metric to evaluate hallucinations in VLMs' answers to visual questions. Researchers have also explored VLMs' ability to abstain when having low confidence to reduce hallucinations (Whitehead et al., 2022; Li et al., 2023b; Wang et al., 2023). We build upon research on automatic evaluation of VQA and further conduct a human evaluation with both sighted and BLV individuals to understand their preferences. As VizWiz frequently contains unanswerable questions, we present an evaluation of VLMs' ability to abstain from answering when provided with images of poor quality.

| | VizWiz | | VizWiz-LF | | | | | | |
|---|---|---|---|---|---|---|---|---|---|
| Category | # of words Question | # of words Answer | Human | GPT-4V | Gemini | # of words LLaVA | QWEN | BLIP-2 | InstructBLIP |
| Identification | 5.2 (4.3) | 1.8 (1.2) | 42.9 (30.0) | **72.0** (25.4) | 15.7 (15.6) | 28.9 (21.0) | 21.2 (19.9) | 12.5 (2.7) | 14.9 (10.8) |
| Description | 7.0 (5.0) | 1.6 (1.4) | 34.8 (30.5) | **57.6** (46.5) | 11.6 (18.0) | 15.0 (17.9) | 13.5 (14.8) | 12.2 (2.7) | 17.4 (13.6) |
| Reading | 7.9 (4.4) | 1.8 (1.9) | 39.8 (36.5) | **66.5** (54.9) | 16.3 (27.6) | 22.2 (30.4) | 18.6 (26.8 | 12.1 (3.0) | 20.2 (17.8) |
| Others | 11.5 (7.4) | 1.6 (1.7) | 47.9 (37.2) | **96.6** (54.1) | 22.3 (29.5) | 41.9 (38.4) | 34.7 (36.2) | 12.6 (3.4) | 23.0 (25.0) |
| Total | 7.9 (5.9) | 1.7 (1.6) | 41.2 (34.0) | **73.2** (48.9) | 16.4 (23.7) | 27.0 (29.8) | 22.0 (26.8) | 12.3 (3.0) | 18.9 (17.9) |

Table 1: Statistics of the sampled VizWiz data and the long-form answers from human expert describers and 6 VLMs. We collect 150 questions from four categories. Numbers in each cell represent the average with the standard deviation in parentheses.

# 3 Dataset

**Visual Questions** Our dataset extends the VizWiz dataset (Gurari et al., 2018) that contains visual questions from BLV individuals seeking visual assistance. To select a diverse range of visual questions, we sample 600 image-question pairs balanced between all four question types (Identification, Description, Reading, and Others) in VizWiz question taxonomy (Brady et al., 2013). See A.1.1 for details on sampling and filtering questions.

**Long-form Answer Collection** To collect long-form answers from expert describers, we hired 20 people experienced in describing images for BLV people using Upwork.[3] Unlike the VizWiz dataset (Gurari et al., 2018) that collected crowd workers' short and quick responses, we encouraged expert describers to write detailed responses. Describers spent 3.9 minutes (SD=1.2) per question (§A.1.2). To evaluate the performance of modern VLMs in LFVQA, we generated long-form answers with six VLMs: GPT-4V (Achiam et al., 2023), Gemini (Team et al., 2023), LLaVA (Liu et al., 2024), InstructBLIP (Dai et al., 2024), Qwen-VL-Chat (Bai et al., 2023), and BLIP-2 (Li et al., 2023a) [4]. We select these models as they are publicly available, have strong zero-shot VQA capabilities, and represent diverse model architectures.

In Table 1, we report statistics comparing the collected long-form answers with the original VizWiz short-form answers [5]. Human expert long-form answers are 24x more words than original VizWiz short answers, indicating previous short answers were insufficient. Modern VLMs also generate long responses, but diverge from human expert answers — GPT-4V answers are 1.7x longer, while the other VLMs generate answers that are 0.7x or shorter.

# 4 Functional Role Analysis

**Functional Roles** We design and annotate functional roles of sentences in LFVQA, to characterize the communicative goal of answer sentences in addressing visual questions. To derive the taxonomy, three authors used open coding (Hashimov, 2015), an iterative process for assigning conceptual labels to segments of data. Then, they met to merge similar functional roles and create a codebook with a name, definition, and example for each functional role. The researchers iteratively coded samples and revised the codebook to achieve final codes, containing eight functional roles (*Confirmation*, *Answer*, *Answer Failure*, *Auxiliary Information*, *Auxiliary Information Failure*, *Explanation*, *Suggestion* and *Miscelleneous*). We provide definitions and examples for each role in §A.2.1.

**Information Source** Short-form answers contain mostly visual information about the image content (Figure 1). We observe that sentences in long-form answers often provide additional information, such as describing the image quality or providing external knowledge outside of the image. Thus, we create a separate taxonomy to categorize sentence-level

---

[3] https://www.upwork.com/

[4] We generated long-form answers with a zero-shot setting. Configuration details in §A.1.3

[5] We used the majority-voted answer among 10 crowd answers (Zeng et al., 2020; Chen et al., 2023a).

| Answers (%) | Confirmation | | Answer | | Ans. Failure | | Auxiliary | | Aux. Failure | | Explanation | | Suggestion | | Misc. | |
|---|---|---|---|---|---|---|---|---|---|---|---|---|---|---|---|---|
| | *ans.* | *unans.* | *ans.* | *unans.* | *ans.* | *unans.* | *ans.* | *unans.* | *ans.* | *unans.* | *ans.* | *unans.* | *ans.* | *unans.* | *ans.* | *unans.* |
| **Expert** | 50 | 50 | 96 | 62 | 6 | 56 | 53 | 45 | 2 | **3** | 35 | 58 | 6 | 27 | 3 | 7 |
| **GPT-4V** | **61** | **60** | 72 | 64 | **38** | **59** | 58 | 52 | **4** | 1 | **69** | **84** | **35** | **63** | 7 | **13** |
| **Gemini** | 29 | 15 | 97 | 87 | 2 | 14 | 16 | 13 | 0 | 0 | 5 | 11 | 2 | 7 | 1 | 3 |
| **LLaVA** | 53 | 39 | **98** | 87 | 2 | 14 | 29 | 29 | 0 | 0 | 11 | 24 | 2 | 15 | 1 | 2 |
| **QWEN** | 46 | 30 | 89 | 62 | 11 | 42 | 17 | 15 | 0 | 1 | 14 | 37 | 2 | 13 | 2 | 6 |
| **BLIP-2** | 46 | 30 | 85 | 76 | 8 | 16 | 3 | 1 | 0 | 0 | 6 | 7 | 1 | 3 | **11** | 11 |
| **InstructBLIP** | 35 | 35 | 96 | **90** | 4 | 10 | 13 | 17 | 0 | 0 | 15 | 17 | 1 | 5 | 2 | 2 |

Table 2: Distribution of answers with each functional role to answerable (*ans.*) and unanswerable (*unans.*) questions. 230 of 600 questions were marked unanswerable in VizWiz.

information sources following the same process. We identify sentences with the following information sources: (1) visual information about the *Image Content*, (2) descriptions of the *Iamage Quality*, and (3) *External Information*. We provide definitions and examples for each information source in §A.2.1.

**Annotation and Classification**  To annotate the functional roles and information source types of the 4.2k long-form answers collected in § 3, we first collect human annotations of 180 long-form answers (522 sentences) as ground-truth. Given the question, the image, and the answer paragraphs, the annotation task is to (1) assign up to three functional roles and (2) assign all applicable information source types for each sentence in the answer paragraph. Five authors participated in three-way annotations and reached substantial agreement with Fleiss Kappa (Fleiss & Cohen, 1973) ($\kappa = 0.76$ for functional roles, $\kappa = 0.87$ for information sources). We annotate the rest of the data using a GPT-4 classifier, which is constructed with a few-shot prompt with definitions of each role and eight in-context examples (see §A.2.2 for the full prompt). We evaluate the classifier's performance against human annotations, achieving a weighted per-role F1 of 0.74 (compared to human-human F1 of 0.79).

## 4.1  Analysis

Table 2-3 shows the distribution of functional roles and information sources of long-form answers in our dataset. As 38% of our sampled questions are marked as unanswerable in VizWiz[6], we report the distribution for answerable and unanswerable questions separately.

**Expert and GPT-4V's answers contain diverse functional roles.**  The most common functional role in answers from both experts and six VLMs was a direct *Answer* to visual questions. LLaVA and Gemini have the highest overall percentage of answers that had *Answer* roles. Gemini often generated single-sentence answers, only addressing the question without providing additional information. On the other hand, expert and GPT-4V answers have sentences with diverse functional roles including *Confirmation*, *Explanation*, *Auxiliary*, and *Suggestion*. GPT-4V answers often contain *Confirmation* sentence when the visual question does not refer to a specific object (*e.g., The image you've provided is blurry, but it shows a part of a bottle with a blue label.*) to confirm that users shared the right image. Answers from both experts and GPT-4V provide *Auxiliary* information to complement the answer. For example, when asked about an attribute of an object in the image (*e.g., "What is the color of this t-shirt?"*), GPT-4V and expert answers additionally describe the same attribute of other objects in the image(*e.g.,* color of the pants), or other attributes of the same object (*e.g.,* text logo on the t-shirt). Example *Misc.* sentences include organization sentences (*e.g., "The screen displays the following text:"*) or final remarks (*e.g., "Hope this helps, thanks!"*) In BLIP-2, answers sometimes repeat the input question and are classified as *Misc*.

**Most VLMs rarely abstain, even when the question is unanswerable.**  Four VLMs (Gemini, LLaVA, BLIP-2, InstructBLIP) all have a low percentage of answers with *Answer Failure* sentences for both answerable and unanswerable questions (Table 2). Expert, GPT-4V, and QWEN indicate *Answer Failure* more often to unanswerable questions, yet GPT-4V over-abstained to answerable questions when the image was not clear. Expert and GPT-4V often

---

[6]Following VizWiz(Gurari et al., 2018), we consider the questions with majority short-form answers of "unanswerable" and "unsuitable" as unanswerable and otherwise answerable.

have both *Answer* and *Answer Failure* sentences in a single answer. These answers typically offer a tentative guess (*e.g.*, *"The color suggests it might be a jar of pasta sauce."*) while also expressing the challenges in providing a definitive answer (*e.g.*, *"However, the text is not completely legible to identify the exact product."*) When GPT-4V abstains from providing an answer to the visual question, the answer often contains an *Explanation* for why it abstained (*e.g.*, describing the low image quality) or a *Suggestion* for taking better images.

**Long-form answers provide information beyond image content.** Table 3 shows the distribution of information source types. Answers from both expert describers and all VLMs have a high percentage of information derived from the *Image Content*. Additionally, expert and GPT-4V answers often describe the *Image Quality*. Compared to other answers, GPT-4V answers provide more *External Information*. For example, when asked to identify the type of frozen food in an image, answers from these models further provide non-visual information, such as the origin of the food and popular recipe used.

| Answers (%) | Image Content | | Image Quality | | External Info. | |
|---|---|---|---|---|---|---|
| | *ans.* | *unans.* | *ans.* | *unans.* | *ans.* | *unans.* |
| **Expert** | **97** | 77 | 24 | 50 | 12 | 25 |
| **GPT-4V** | 86 | 82 | **56** | **67** | **45** | **55** |
| **Gemini** | 90 | **84** | 5 | 6 | 16 | 24 |
| **LLaVA** | 95 | 78 | 2 | 7 | 15 | 33 |
| **QWEN** | 86 | 60 | 5 | 7 | 8 | 20 |
| **BLIP-2** | 84 | 60 | 2 | 3 | 6 | 17 |
| **InstructBLIP** | 84 | 68 | 5 | 4 | 12 | 35 |

Table 3: Distribution of answers with each information sources to answerable (*ans.*) and unanswerable (*unans.*) questions. Among 600 questions, 230 were marked unanswerable in VizWiz.

## 5 Automatic Evaluation

To analyze the performance of VLMs in LFVQA, we conduct an automatic evaluation of long-form answers with reference-based metrics. To adapt the traditional reference-based metrics to long-form answers, we consider both short crowd answers (Gurari et al., 2018) and long expert answers (our dataset) as ground truth references and explore how our functional role classifier can be used for sentence-level evaluation. To our knowledge, we are the first to consider long-form reference answers when evaluating VQA.

**Data** We selected 360 examples in our dataset written by experts, GPT-4V and Gemini (demonstrated as state-of-the-art for image understanding tasks (Yue et al., 2024; Team et al., 2023) and used in accessibility apps (BeMyEyes, 2020; TalkBack, 2024)), given the cost of evaluation. The samples were balanced across question categories (Identification, Description, Reading, and Others).

**Method** We evaluate long-form answers using four reference-based VQA evaluation metrics: ROUGE (Lin, 2004), METEOR (Elliott & Keller, 2013), BERTScore (Zhang et al., 2019), and LAVE, an LLM-based metric that uses GPT-4 (Mañas et al., 2023) to compare a candidate answer against a reference answer (details in §A.3.1). Comparing long-form answers to short-form reference answers can penalize long-form answers for including additional information (*e.g.*, explanation, suggestion) (Krishna et al., 2021; Mañas et al., 2023). To better evaluate LFVQA, we consider two approaches. First, we explore the potential of long-form references in VQA evaluation by leveraging the expert long-form answers in our *VizWiz-LF* as ground truths ($l_r$) for evaluating model long answers ($l_c$). However, many existing VQA datasets have only short ground truth answers (Antol et al., 2015; Gurari et al., 2018), and collecting long-form ground truths on a large scale is costly. Thus, we also explore the use of functional roles in VQA evaluation with short-form references ($s_r$) as ground truths for evaluating extracted *Answer* role sentences from long-form answers ($l'_c$).

**Results** Table 4 summarizes the evaluation results with reference-based metrics. In conventional reference-based metrics (ROUGE, METEOR, BERTScore), Gemini outperformed GPT-4V and experts. This was often due to the additional information in the long-form answers of GPT-4V and experts being penalized by reference-based metrics. In LLM-based

| Answer Source | Lex-Overlap (Unigram) | ROUGE | | | | METEOR | | | | BERTScore | | | | GPT-4 | | | |
|---|---|---|---|---|---|---|---|---|---|---|---|---|---|---|---|---|---|
| | | $s_r+l_c$ | $s_r+l'_c$ | $l_r+l_c$ | $l_r+l'_c$ | $s_r+l_c$ | $s_r+l'_c$ | $l_r+l_c$ | $l_r+l'_c$ | $s_r+l_c$ | $s_r+l'_c$ | $l_r+l_c$ | $l_r+l'_c$ | $s_r+l_c$ | $s_r+l'_c$ | $l_r+l_c$ | $l_r+l'_c$ |
| **GPT-4V** | 0.35 | 0.04 | 0.19 | **0.22** | **0.32** | 0.08 | 0.17 | **0.29** | **0.31** | 0.8 | 0.83 | **0.87** | **0.88** | **0.76** | **0.76** | **0.76** | **0.73** |
| **Gemini** | 0.22 | **0.19** | **0.22** | **0.22** | 0.26 | **0.18** | **0.20** | 0.16 | 0.21 | **0.84** | **0.84** | **0.87** | 0.87 | 0.63 | 0.64 | 0.45 | 0.51 |
| **Expert** | **0.36** | 0.06 | 0.2 | - | - | 0.09 | 0.18 | - | - | 0.81 | **0.84** | - | - | 0.71 | 0.7 | - | - |

Table 4: Evaluation results with reference-based metrics (*reference+candidate*). For reference answers, we use VizWiz crowd's majority-voted answers ($s_r$), experts' long-form answers ($l_r$), and extracted answer sentences of experts' long-form answers ($l'_r$). For candidate answers, we use original long-form answers generated by models and experts ($l_c$) and extracted answer sentences ($l'_c$).

evaluation (LAVE), the trend reversed, and GPT-4V and experts scored higher than Gemini. Among all metrics, only the LLM-based metric demonstrated a moderate correlation to human ratings (§A.3.1), demonstrating the potential of LLMs in LFVQA evaluations. For all metrics, using long-form answers as reference answers increased the score compared to using short reference answers ($s_r$). GPT-4V answers that showed low scores with $s_r$ exhibited a more substantial improvement in score for ROUGE and METEOR with long-form references ($l_r$) (Figure 6 in Appendix). While expert long-form answers are difficult to collect at scale, our evaluation highlights the need for long-form references to effectively evaluate LFVQA. Also, future work can consider more diverse functional roles than *answer* roles to enable a more fine-grained evaluation of VQA. For instance, sentences with *confirmation of photo* roles can be evaluated using image groundings, and *external information* sentences can be assessed with fact-checking approaches.

## 6   Human Evaluation

While prior research with BLV people has shown that they want detailed image descriptions (MacLeod et al., 2017; Salisbury et al., 2017), no work has explored BLV people's preferences for answer lengths in VQA. Thus, we first conduct a preliminary survey with 8 BLV people who compare short answers from crowd workers (from VizWiz) vs long answers from expert describers (from VizWiz-LF). Participants preferred long-form answers 75% more often than short-form answers, and this preference was significant ($p < .0001$, details in §A.3.2). Participants ranked shorter answers higher than long-form answers when the question asked about a specific attribute of an object (*e.g., "What is the color of this t-shirt?"*).

To understand preferences for LFVQA, we conduct an evaluation study with 20 BLV and 20 sighted people [7] While most prior research in VQA evaluates answers with sighted people (Dua et al., 2021; Gardner et al., 2020), long-form answers should be evaluated beyond factual accuracy to consider their usefulness to end users, specifically BLV people. We conduct evaluations under all conditions of {BLV, sighted}×{with image (description), without image (description)} to account for both scenarios where users have or do not have the context regarding the image settings. For BLV participants, we share a brief image description to give them context for evaluating the answer. We obtain the image description from the original VizWiz dataset (Gurari et al., 2018) authored by crowd workers [8].

**Method**   We conducted a human evaluation on the same data used in Section 5. The evaluation study involved (1) a preference ranking task and (2) a fine-grained answer rating task followed by an open-ended interview to understand their rankings and ratings. In the first task, evaluators were provided 12 visual questions, 3 from each question category (*Identification, Description, Reading,* and *Others*). For each visual question, participants read and ranked long-form answers from 3 sources (GPT-4V, Gemini, and expert describers) for a total of 36 long-form answers. In the second task, evaluators provided a set of ratings for

---

[7]Approved by our Institutional Review Board (IRB). Demographics of BLV participants in §A.3.3.
[8]We randomly sample one from the five default captions after manually removing spam captions.

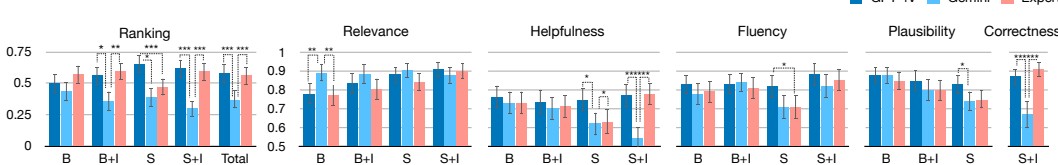

Figure 2: Average rankings and ratings of long-form answers (GPT-4V, Gemini, Expert) across four groups: BLV (*B*), BLV with image caption (*B+I*), Sighted without image (*S*), Sighted with image (*S+I*). We measured significance using the Freidman test followed by the pair-wise Wilcoxon test. $p < 0.05$ marked with *, $p < 0.01$ marked with *, $p < 0.001$ marked with *** after Bonferroni correction.

36 long-form answers.[9]. Participants provided one of the three labels (1 - not, 2 - partially, and 3 - very) for each criterion:

- *Relevance* measures how relevant the answer is to the question (Levinboim et al., 2019). It is important to understand how people rate relevance in long-form answers as they often contain extra information in addition to the core answer.
- *Helpfulness* measures how helpful the answer is for people who cannot see the image.
- *Plausibility* measures how likely the answer is to be correct (Gardner et al., 2020).
- *Fluency* measures how clearly the response conveys information, often related to the answer's grammar or consistency (Levinboim et al., 2019).

For the group of sighted evaluators who were provided with images during the evaluation, they are asked to evalaute with *Correctness* (*i.e.,* how correct the provided information is given the image) instead of *Plausibility*.

**Results**   Figure 2 displays the normalized rankings and ratings of three long-form answer types across all four participant groups. All evaluator groups ranked responses from experts and GPT-4V higher than those from Gemini, often attributing their choice to the rich information in expert and GPT-4V responses. For relevance specifically, BLV evaluators preferred shorter answers from Gemini as they found GPT-4V answers often share extra information that is not useful (*e.g.,* describing a shirt's surroundings when asked for the shirt's color). BLV evaluators also noted that GPT-4V's frequent image quality descriptions (*e.g.,* "You uploaded a blurry photo of...") felt repetitive and unnecessary, unless used to explain an answer failure, as they often took low-quality photos.

Sighted evaluators assessing the answer with the image (S+I in Figure 2) rated Gemini answers significantly lower than GPT-4V and expert answers for *Helpfulness* and *Correctness* and explained that low ratings were due to incorrect information provided in the answer. In contrast, BLV groups rated *Plausibility* high (0.84 out of 1.0, SD=0.28) for all types of long-form answers. Our follow-up interviews with BLV evaluators revealed that their trust in long-form answers is influenced by the level of detail and expressions of uncertainty in the answer. For GPT-4V and expert answers, extensive details made BLV evaluators perceive the answers as more correct. However, compared to Gemini, GPT-4V and experts frequently expressed uncertainty due to low image quality, thus BLV evaluators trusted Gemini answers despite their brevity.

## 7   Experiments on VQA Model Abstention

Our human evaluation reveals that the perceived correctness of long-form answers for BLV raters does not align with the answer correctness. Assessing answer correctness is challenging for BLV users, as they cannot visually verify the answer against the image [10]. Prior work

---

[9]We sample one answer from each visual question to avoid bias (*e.g.,* giving high ratings if two answers provide the same information.)

[10]As a workaround, BLV people check for common object detection errors Huh et al. (2023b) or conflicting information among multiple model outputs Huh et al. (2023a).

| Model | abs-acc | | abs-prec | | abs-recall | | abs-f1 | | abs-% | | ans-acc | |
|---|---|---|---|---|---|---|---|---|---|---|---|---|
| | vanilla | prompt | vanilla | prompt | vanilla | prompt | vanilla | prompt | vanilla | prompt | vanilla | prompt |
| **Random** | 0.50 | - | 0.34 | - | 0.34 | - | 0.34 | - | 38 | - | - | - |
| **Majority** | 0.62 | - | 0.0 | - | 0.0 | - | 0.0 | - | 0 | - | - | - |
| **Expert** | 0.73 | - | 0.68 | - | 0.57 | - | 0.62 | - | 32 | - | 0.60 | - |
| **GPT-4V** | **0.77** | 0.73 | 0.66 | 0.60 | **0.82** | 0.88 | **0.73** | 0.71 | 47 | 56 | **0.68** | 0.68 |
| **Gemini** | 0.66 | 0.71 | 0.78 | 0.65 | 0.14 | 0.51 | 0.24 | 0.57 | 7 | 30 | 0.46 | 0.55 |
| **LLaVA** | 0.66 | **0.78** | **0.80** | 0.70 | 0.14 | 0.74 | 0.24 | **0.72** | 7 | 41 | 0.43 | 0.61 |
| **QWEN** | 0.71 | 0.76 | 0.70 | 0.64 | 0.42 | 0.84 | 0.52 | **0.72** | 23 | 50 | 0.52 | 0.64 |
| **BLIP-2** | 0.63 | 0.70 | 0.54 | 0.59 | 0.16 | 0.73 | 0.24 | 0.65 | 11 | 47 | 0.27 | 0.54 |
| **InstBLIP** | 0.63 | 0.65 | 0.63 | 0.53 | 0.10 | 0.72 | 0.17 | 0.61 | 6 | 52 | 0.25 | 0.41 |

Table 5: Performance of vanilla model and abstention instruction on six VLMs. Full results with other abstention strategies in Table 26 in the Appendix.

shows that BLV people often fill in details to resolve incongruent model descriptions rather than suspecting errors (MacLeod et al., 2017). Thus, false information hallucinated by VLMs can mislead users. Unanswerable questions are prevalent in questions asked by BLV users due to poor image conditions (*e.g.,* blurry images, absence of target object), yet VLMs often hallucinate answers to such questions §5. We quantify the level of hallucinations of VLMs with such images and evaluate their ability to correctly abstain from providing an answer when the question is unanswerable  (Xiong et al., 2023; Feng et al., 2024).[11] We investigate a suite of prompting methods and evaluate their effectiveness on our dataset.

## 7.1   Experiment Setting

**Data and Answerability Labels**   We consider the 600 VizWiz questions in our dataset (§3) and use short-form answers from VizWiz as gold answerability labels, for a total of 38% of questions labeled unanswerable. To determine whether a long-form answer abstains from providing an actual answer, we classify its sentence-level functional roles (§4) and check whether at least one sentence is labeled as *Answer Failure*, which expresses a model's inability to answer the question (examples in Table 21).

**Evaluation Metrics**   We report four sets of metrics: (1) Abstain Accuracy (**abs-acc**), which measures whether the abstain decisions are correct against the gold answerability label; (2) Abstain Precision (**abs-prec**), Recall (**abs-recall**) and F1 (**abs-f1**), treating abstaining to answer as the positive label; (3) Percentage of questions that the model abstains from answering (**abs-%**) and (4) Answer Accuracy (**ans-acc**), which compares the answer against the ground truth short answers using LLM-based metric (Mañas et al., 2023), as it correlates well with human judgment (§6). We report the performance of three baselines: a **random** where an answerability label is randomly assigned according to the distribution and a **majority** baseline where we always predict *Answerable*. We also report **expert** performance by evaluating the long-form answers written by experts.

**Abstention Strategies**   We investigate VQA models' abstention capabilities using three prompting techniques. The first approach *Abstention Instruction* includes a detailed instruction in the prompt and instructs the model to determine whether a question is unanswerable (*e.g.*, overly blurry images), and if so, abstain from answering. We also used two additional prompting strategies from prior work Cui et al. (2023); Wang et al. (2022) but they do not yield performance gains (details in §A.4.2).

## 7.2   Results

Table 5 presents the performance of the default (*vanilla*) and abstention instruction (*prompt*) approaches. The abstention instruction prompt increases abstention frequency across all models (*abs-%*), while also improving the ability of all models to correctly abstain from

---

[11]We do not evaluate the factuality of *all* the information presented in the long-form answer, which might also contain hallucinations. Future work can leverage our role analysis to identify appropriate sources for the factuality evaluation of the entire answer.

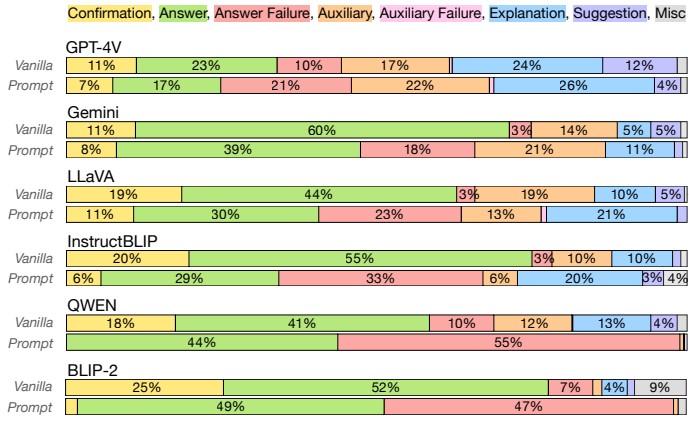

Figure 3: Functional role distribution of 6 models' long-form answers before and after applying abstention instruction.

answering unanswerable questions (*abs-recall*), especially for models with low abstention frequency in the *vanilla* setting (Gemini, LLaVA, BLIP-2, InstructBLIP). While precision (*abs-prec*) declines with abstention instructions, the F1 (*abs-f1*) and answer accuracy (*ans-acc*) increase for all models except GPT-4V. For low-quality images, opting to abstain rather than providing low-confidence answers may benefit users in the context of accessibility where wrong but confident answers can cause harm.

Prompting VQA models to abstain also changes the functional role distributions in their answers compared to the *vanilla* setting (Figure 3). Answers from GPT-4V, Gemini, LLaVA, and InstructBLIP provide more auxiliary information and explanations when instructed to abstain, whereas answers from QWEN and BLIP-2 exhibit patterns of short-form VQA by generating primarily answer or answer failure roles.

Additionally, we note a discrepancy in abstention between expert answers and crowd-sourced short answers. Expert and crowd answers agreed on answerability in 52% of the 600 questions. Among 230 questions marked unanswerable by the crowd workers, 99 were answered by expert describers. This often occurred when a few crowd workers provided an answer but the question was labeled "unanswerable" using majority voting (§ A.4.3). Among 370 questions marked answerable by VizWiz, 61 were not answered by experts. This occurred when the experts were not familiar with specific objects in *Identification* questions.

## 8   Conclusion

In this work, we presented *VizWiz-LF*, a dataset of long-form answers to visual questions posed by BLV users. Our work is the first to explore the content and structure of LFVQA from both humans and models, and to compare long-form answers to short-form answers. Our functional role and information source analysis demonstrates the rich information provided in LFVQA by both human experts and models. We also revealed that while BLV evaluators preferred long-form answers to short-form answers, there remain opportunities to improve the relevance and correctness of LFVQA. We found that abstention instructions helped VQA models to better abstain from answering unanswerable questions. Future work may use functional roles to further evaluate and improve the per-role performance of LFVQA and to tailor descriptions to context and preference. In this era of VQA, we must go beyond domain-agnostic factual accuracy to consider how to efficiently deliver relevant information. We hope that our work encourages future development and evaluation of VLMs with people who may benefit the most from using them.

Acknowledgments

The study is partially funded by NSF grant IIS-2312948.

**Ethics Statement**

- **Hallucinations in LLMs and VLMs:** LFVQA answers in our dataset are collected from expert describers and VLMs. As LLMs and VLMs might generate factual errors and hallucinations, we have examined the data to make sure they do not contain harmful content before the human evaluation study. Our work also uses LLMs to classify functional roles and evaluate VQA. While we have shown that these approaches better align with humans, LLMs are not free from potential hallucinations and can lead to incorrect results.

- **Human Evaluation Study:** Our human evaluation study was approved by our institution's International Review Board (IRB) as exempt research. In our user study, we ensured that all participants were compensated fairly for their time and contributions. The payment was determined based on the average market rate for such studies, reflecting the complexity and duration of the tasks involved.

- **Application to other domains:** The experience of visual disabilities is individual and can affect people's preferences in image description (Stangl et al., 2021; Huh et al., 2022). With our proposed functional roles of LFVQA, future systems can support VQA of adaptive length and content based on user contexts (Kreiss et al., 2022; Peng et al., 2022; Gubbi Mohanbabu & Pavel, 2024; Srivatsan et al., 2023) and preferences (Jones et al., 2024; Van Daele et al., 2024). While we analyzed long-form answers to visual questions asked by BLV people, LFVQA also occurs in other use cases (*e.g.*, asking how to solve a math problem with a screenshot of the problem) (Chen et al., 2023b). We expect that our functional roles will cover common roles and information types in other domains (*e.g.*, answer, auxiliary information, suggestion). Other domains are likely to encounter different proportions of our functional roles and information types (*e.g.*, fewer "image quality" issues and related suggestions) and other domains may have additional functional roles and information types we have not yet observed. Future work will explore extending our functional roles and information types to other domains.

- **Risk of model evaluation for accessibility use cases:** Unlike applications of VQA where people can visually check the answers to their questions, BLV people consume model answers without further verification. Our human evaluation study reveals that BLV people have higher trust in model-generated answers than sighted people. Additionally, high scores from evaluations may further encourage trust in models leading to potential risks of unnoticed model errors. We hope robust evaluation approaches and an understanding of long-form answers can provide a more realistic picture of model performance.

**Reproducibility Statement**  We will release our codes, prompt, and data collected publicly.

- **Collecting Long-form Expert Descriptions:** Our *VizWiz-LF* dataset contains long-form answers from expert describers who are skilled in describing images for BLV people. Because what they decide to include in the long-form answers can be subjective, the diversity and representativeness of the human describers can influence the results. Also, compared to hiring crowd workers, it is more challenging to scale up the collection of expert long-form answers due to the difficulty of finding skilled describers and the cost. We collected one expert long-form answer per visual question to prioritize a larger dataset over multiple long-form answers for a smaller dataset which aligns with prior work (Gurari et al., 2019; Zhou et al., 2017) that collects one annotation per example from expert annotators. Future work can augment our VizWiz-LF dataset with more ground truth references and perform human correlation studies (Gupta et al., 2019).

- **Use of LLM as an evaluator:** We use GPT-4 in classifying functional roles and evaluating answer accuracies. While prior work has demonstrated its improved correlation with human judgment (Mañas et al., 2023; Chan et al., 2023), we acknowledge that results are nondeterministic. Additionally, as some of the LLMs and VLMs explored in our paper are not open-source, the models are subject to arbitrary change, replacement, or removal. We hope that efforts to increase open-access language models will alleviate these concerns in the future. Finally, our work relies on language models that contain many billions of parameters and involve expensive API calls. We hope that future advances in LLM inference and architecture will enable low-cost use of these models with good performance.

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

# A  Appendix

## A.1  Dataset Collection

### A.1.1  Sampling VizWiz Questions

We first randomly selected 2,500 unique image-question pairs from VizWiz's training and validation sets as an initial pool to further sample diverse types of questions. We removed image-question pairs with incomplete questions (*e.g., Is this?*) and questions about the VizWiz service (*e.g., Is there someone answering questions 24 hours a day?*), resulting in 2447 total pairs.

To select a diverse range of image-question pairs, we categorized the 2,500 questions into four question types (Identification, Description, Reading, and Others) from an existing VizWiz question taxonomy (Brady et al., 2013), then randomly sampled 150 image-question pairs from each of the four question types to obtain a total of 600 image-question pairs. In *Identification* questions, users ask to identify an object. In *Description* questions, users ask about the visual attributes (*e.g.,* color, count, style) of an object or setting. In *Reading* questions, users ask for the text to be read. *Others* category includes examples with multiple questions from different categories, or questions that involve further reasoning or knowledge outside the image. We did not remove low-quality images (*e.g.,* blurry, dark) or image-question pairs labeled "unanswerable" by crowd workers as we aimed to capture expert and vision language model responses to these scenarios.

### A.1.2  Expert Answer Collection

To collect long-form answers from expert describers, we hired 20 people experienced in describing images for BLV people using Upwork (2024). While VizWiz crowd workers aimed to answer the questions in nearly real-time (36 sec, SD=30), we encouraged expert describers to write ideal and detailed responses (234 sec, SD=1.17). We paid expert described by their hourly rate (28 USD, SD=7.84). High-quality annotations are exemplified in Figure 4.

### A.1.3  Model Answer Collection

Table 6 provides architecture configurations of the four benchmarked Vision-Language Models (VLMs), including specifications of their language encoders, vision encoders, and adapters. GPT-4V and Gemini's architectures are undisclosed. We use the default VLM temperatures (gpt-4-1106-vision-preview: 1.0, gemini-1.0-pro-vision: 0.4, llava-v1.5-13b: 0, blip2-flan-t5-xxl: 1, instructblip-flan-t5-xxl: 1) and top-p (qwen-vl-chat: 0.3, instructblip-flan-t5-xxl: 1).

| Model | Language Encoder | Vision Encoder | Adapter |
|---|---|---|---|
| **LLaVA-1.5** | Vicuna-13B | ViT-L/14 Zhai et al. (2022) | FC Layer |
| **BLIP-2** | FlanT5-XXL(11B) (Chung et al., 2022) | ViT-g/14 (Fang et al., 2023) | Q-Former |
| **QWEN-VL-Chat** | Qwen-7B | ViT-bigG (Gadre et al., 2024) | Position-aware Cross-Attention Module |
| **InstructBLIP** | FlanT5-XXL(11B) (Chung et al., 2022) | ViT-g/14 (Fang et al., 2023) | Q-Former |

Table 6: Details about the four benchmarked VLMs' model architecture.

## A.1.4 Examples of Long-form Answers in *VizWiz-LF* dataset.

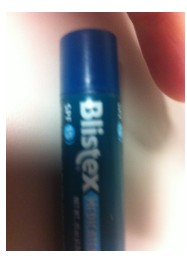

**Question**
*What is this?*

**VizWiz original**
*blistex*

**GPT-4V**
*The item in the image is a tube of Blistex lip balm. Blistex is a brand that produces lip care products, and this particular tube seems to include an SPF, indicating that it offers sun protection as well.*

**Gemini**
*This is a tube of Blistex lip balm.*

**Expert**
*The photo shows a tube of deep teal colored Blistex. Other contents seen is the word: "medicated" and a SPF label but the number is not clear.*

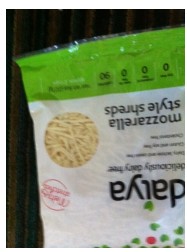

**Question**
*What's the expiration date on this non dairy cheese?*

**VizWiz original**
*unanswerable*

**GPT-4V**
*I'm sorry, but the image is too blurry for me to accurately read the expiration date on the non-dairy cheese package. If you can provide a clearer image or the printed date, I'd be happy to help you determine the expiration date.*

**Gemini**
*06 05 2023*

**Expert**
*The expiration date on the cheese is not visible in the image.*

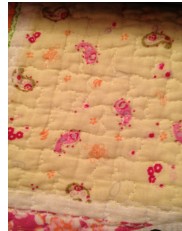

**Question**
*What is that color?*

**VizWiz original**
*yellow*

**GPT-4V**
*The image shows a fabric with a light yellow or cream base color, adorned with patterns in various colors, including pink, purple, and orange. The patterns consist of floral and paisley-like designs. Please note that the perception of color can vary depending on the lighting conditions and the settings of the device displaying the image.*

**Gemini**
*Pale yellow*

**Expert**
*The color of the bedsheet is soft orange. It also has different designs and flowers in pink, yellow and red colors.*

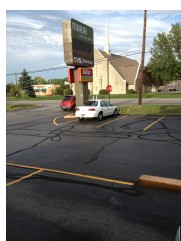

**Question**
*What's the make and model of that car, and what's the license plate number?*

**VizWiz original**
*blistex*

**GPT-4V**
*The car in the photo appears to be a Chevrolet Corvette, likely a model from the late '90s or early 2000s, given its design characteristics. As for the license plate, I cannot provide that information. License plates are personal data specific to an individual's vehicle registration and should not be shared online for privacy reasons.*

**Gemini**
*The make and model of the car is a Chevrolet Cavalier. The license plate number is 367WMD.*

**Expert**
*The car is a white Chrysler mid-size car. The model is not visible. The license plate consists of white numbers on a blue background. The license plate is blurry but appears to consist of four numbers.*

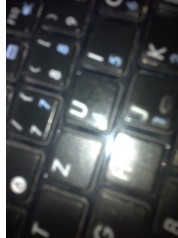

**Question**
*What is this?*

**VizWiz original**
*keyboard*

**GPT-4V**
*This appears to be a close-up, blurry photograph of a keyboard, with individual keycaps showing characters or letters. The photo is taken from such an angle and with such lighting that there's a bright glare reflecting off the keys, which makes it a bit challenging to see the details clearly. The keyboard looks like it might be used with a desktop computer or a laptop.*

**Gemini**
*It's a laptop keyboard.*

**Expert**
*The image is a blurry close up of a keyboard with some reflected light in the middle of the image.*

Figure 4: Examples of Long-form Answers in *VizWiz-LF* dataset. Images, questions, and short-form answers from VizWiz-VQA dataset

A.2   Functional Roles and Information Types

A.2.1   Human Annotaions

To derive the taxonomy, three researchers used open coding on 100 sample responses to obtain potential functional roles. Then, they met to merge together similar functional roles and create a codebook with a name, definition, and example for each functional role. The researchers iteratively coded samples and revised the codebook to achieve final codes, containing eight functional roles. During the coding process, an additional need for annotating information types emerged. To better understand long-form answers that provide information beyond the image content, the researchers additionally generated the codebook for information source types. The full codebook and examples provided to annotators are shown in Table 7.

Using the codebook with examples, we collected three-way annotations of 180 long-form answers from five researchers. They reached a substantial agreement for functional roles (Fleiss Kappa = 0.76) and a perfect agreement for information types (Fleiss Kappa = 0.81). These high-quality annotations were used as few-shot examples for prompting a classifier and to evaluate their performance.

| Role | Definition | Example |
|---|---|---|
| **Confirmation** | Confirms what the user uploaded. | *The image you uploaded appears to be a carton of chocolate soymilk.* |
| **Answer** | Addresses the question with an answer. | *It expires in September 2015.* |
| **Answer (Failure)** | States the inability to address the question. | *The expiration date is not visible in this photo.* |
| **Auxiliary** | Provides additional information not directly related to the question. | *The size of the milk container is 8 fluid ounces (240 mL).* |
| **Auxiliary (Failure)** | States the inability to provide information not directly related to the question. | *I cannot provide nutritional information due to the blur.* |
| **Explanation** | The sentence explains the reasoning for the information it provides. | *Given the presence of soybeans in the image, it is likely to be soymilk.* |
| **Suggestion** | Suggests retaking or improving the quality of a photo to get a better answer. | *If you can provide a clearer image, I might be able to better assist.* |
| **Miscelleneous** | Does not provide new information. Sentences that do not belong in any of the categories above. | *I'm happy to assist you, let me know if you have further requests!* |

Table 7: Definitions of functional roles identified in VizWiz-LF dataset

| Type | Definition | Example |
|---|---|---|
| **Image Content** | Provides visual information about the image content | *You are holding a pink bottle.* |
| **Image Quality** | Provides visual information about the image quality. | *The image you've provided is quite blurry and the details are not clear* |
| **External Information** | Provides general knowledge or facts that are not found in the image. | *Expiration date is typically located near the top or side of the bottle* |

Table 8: Definitions of information sources identified in VizWiz-LF dataset

A.2.2   Classifier

We construct a prompt with definitions and in-context examples for each of the function roles or information types. Given a question and a list of answer sentences, GPT-4 outputs a list of labels for each of the answer sentences. The prompt (which includes the task instruction and few-shot examples) we use can be found in Table 11- 12 and Table 13. We

evaluate GPT-4's performance against the majority label from the three-way annotations, using per-label F1 and a weighted average F1 over all the class.

To contextualize GPT-4's performance, we provide two estimates for human performance collected in §A.2.1: an **upperbound**, which we compare each annotator's annotation with the majority label. This inflates the performance as one's annotation is correlated with the majority label; an **lowerbound** which we compare all pairs of annotation and average over them. We report two baselines: (1) **Random**: which randomly assigns a role combination from all of the annotation and (2) **Majority**: which always labels the sentence as "Answer" (or "Image Content" for information type).

Results are in Table 9 and Table10. We see that GPT-4 significantly outperforms both the baselines, with a moderate gap compared to the human lowerbound. The model performs relatively poorly in identifying "Auxiliary (Failure)" and "Confirmation", for which human also exhibit lower agreements.

| Model | Confirmation | Answer | Ans. Failure | Auxiliary | Aux. Failure | Explanation | Suggestion | Misc. | Average |
|---|---|---|---|---|---|---|---|---|---|
| **Majority** | 0.0 | 0.74 | 0.0 | 0.0 | 0.0 | 0.0 | 0.0 | 0.0 | 0.35 |
| **Random** | 0.13 | 0.57 | 0.04 | 0.29 | 0.0 | 0.01 | 0.13 | 0.0 | 0.37 |
| **GPT-4 (8-shot)** | 0.40 | 0.84 | 0.79 | 0.65 | 0.18 | 0.60 | 0.86 | 0.53 | 0.74 |
| **Human (lower)** | 0.51 | 0.89 | 0.80 | 0.69 | 0.43 | 0.70 | 0.80 | 0.79 | 0.79 |
| **Human (upper)** | 0.75 | 0.95 | 0.90 | 0.85 | 0.67 | 0.86 | 0.90 | 0.90 | 0.89 |

Table 9: Per-role F-1 for automatic functional role classification.

| Model | Image Content | Image Quality | External Info. |
|---|---|---|---|
| **Majority** | 0.91 | 0.0 | 0.0 |
| **Random** | 0.82 | 0.08 | 0.05 |
| **GPT-4 (8-shot)** | 0.95 | 0.74 | 0.77 |
| **GPT-4V** | 0.78 | 0.72 | 0.72 |
| **Human (lower)** | 0.97 | 0.92 | 0.88 |
| **Human (upper)** | 0.95 | 0.83 | 0.77 |

Table 10: Per-role F-1 for automatic information source classification.

You are given a question to an image and an answer paragraph to the question. You job is to assign each of the sentence in the answer paragraph into at least one and up to three functional roles listed below. For each sentence, please assign all the roles that are applicable.

# Role: Confirmation of Photo
# Definition: The sentence confirms what the user uploaded. When the user asks an identification question (e.g., what is this?), this sentence may also be annotated as an answer.
*This usually comes at the beginning of the sentence, to provide the overview of the image. This sentence often looks similar to typical image captions.
# Example: "The image you've provided appears to be of a SodaStream Raspberry flavor syrup bottle."

# Role: Answer
# Definition: The sentence directly addresses the question. If the answer is provided in multiple sentences, they can all be labeled as "answer" as in the example below. Incorrect answers are still labeled as answers.
# Example: (question: what color is this?) "I would describe the shirt as a reddish-brown color."

# Role: Answer Failure
# Definition: The sentence states the inability to address the question, often accompanied by an "Explanation of Reasoning" explaining the reason.
# Example: "I cannot provide information such as the details on this globe."

# Role: Auxiliary
# Definition: The sentence provides additional information not directly related to the question but observed in the photo, or general knowledge or facts related to the query.
# Example: (question: Read this label) "Pectin is a natural thickening agent that's extracted from fruits and used commonly in cooking to gel liquids."

# Role: Auxiliary Failure
# Definition: The sentence states the inability to provide auxiliary information not directly related to the question.
# Example: (question: What is the color of the pants?) "However, I cannot identify its brand name because the image is blurry."

# Role: Explanation of Reasoning
# Definition: The sentence explains the reasoning for the information it gives by describing its thought process or providing the information source. It can also explain the reasoning behind failure (e.g., low quality of the photo).
# Example: (question: what is this?) "Given the color of the liquid visible through the bottle, it's likely to be a type of mustard sauce, possibly a honey mustard dressing or similar condiment based on its golden-brown hue."

# Role: Suggestion
# Definition: The sentence suggests users retake the photo, provide additional information, or try a different method or strategy to find the answer. It may also advise caution or exceptions.
# Example: "If you need the exact instructions, you might try using a smartphone to take a clearer picture of the instructions on the box."

# Role: Misc
# Definition: The sentence does not provide new information. Examples can be sentences that confirm what the user asked, organizational sentences (explaining how the answer will be structured before elaborating), and final remarks. Any sentences that do not belong in any of the categories above.
# Example: "I'm happy to assist you, please let me know if you have any other requests!"

Question: How many video games are in these shelves?
Answer:
[0] The image shows a shelf with a collection of video games.
[1] The exact number of games is not fully discernible due to the angle and focus of the photo, but I will do my best to estimate based on what is visible.
[2] The top shelf appears to have around 11 games visible, the middle shelf has approximately 30, and the bottom shelf has around 27.
[3] Please note that these numbers are rough estimates as the photo does not clearly show all games, and there may be additional games stacked or hidden behind the ones in front.
[4] Therefore, we can approximate that there are at least 68 video games visible in the shelves.
[5] However, the actual number could be slightly higher if there are multiple layers of games or games obscured from view.
Roles:
[0] Confirmation of Photo
[1] Answer Failure, Explanation of Reasoning, Misc
[2] Answer, Explanation of Reasoning
[3] Suggestion, Explanation of Reasoning
[4] Answer
[5] Suggestion, Explanation of Reasoning

Table 11: Prompt for functional role classification. Continued in Table 12.

Question: What this?
Answer:
[0] The image you've provided is blurry, but it shows a part of a bottle with a blue label.
[1] There appears to be a barcode and possibly a brand logo, but due to the quality of the image, it's not possible to identify the contents of the bottle with certainty.
[2] If you have more information or a clearer picture, I could attempt to provide a more accurate identification.
Roles:
[0] Confirmation of Photo, Answer
[1] Answer Failure, Explanation of Reasoning
[2] Suggestion

Question: What is in this bottle?
Answer:
[0] The image is blurry, but it shows a bottle with a label that includes the word "MUSTARD".
[1] Given the color of the liquid visible through the bottle, it's likely to be a type of mustard sauce, possibly a honey mustard dressing or similar condiment based on its golden-brown hue.
[2] Mustard sauces are commonly used as a condiment or dressing for sandwiches, salads, and many other dishes.
Roles:
[0] Confirmation of Photo, Auxiliary
[1] Answer, Explanation of Reasoning
[2] Auxiliary

Question: What's in this?
Answer:
[0] This image is very close to the product and a little blurry.
[1] It shows goldfish crackers inside a clear ziploc container top and the bottom looks light blue holder.
[2] It shows goldfish crackers inside a clear ziploc container top and the bottom looks light blue holder.
Roles:
[0] Auxiliary
[1] Answer
[2] Auxiliary

Question: Do this tie and shirt match?
Answer:
[0] No, the tie and shirt do not match.
[1] The shirt is a light green and white striped pattern, while the tie is a dark blue with light blue and white squares.
Roles:
[0] Answer
[1] Auxiliary, Explanation of Reasoning

Question: What's this? Can you describe?
Answer:
[0] Shown in the photo is a tabletop with a Braille display/keyboard, several wires and charging chords, and a small bit of blue fabric, possible a bag or clothing.
[1] The make and model of the display is not visible but the keyboard is a matte Silver/grey.
[2] At the top are the braile input buttons, beneath which is the tactile readout.
[3] Jutting from the bottom, taking up the middle 1/3 of the display is another row of buttons, the center of which apears to be a directional pad.
Roles:
[0] Confirmation of Photo, Answer
[1] Auxiliary, Auxiliary Failure
[2] Auxiliary
[3] Auxiliary

Question: Hi, I'm going to start over. This is the place mat that are related to the napkins. First, I need to know if the stripes are vertical or horizontal.
Answer:
[0] The stripes on the place mat are horizontal, meaning they run from left to right across the width of the place mat.
Roles:
[0] Answer, Explanation of Reasoning

Question: Hey can you tell me how much this weighs? Thanks.
Answer:
[0] No weight visible or discoverable by Google search.
[1] Kroger home sense white unscented facial tissues, 15 pack of 2 ply facial tissues.
Roles:
[0] Answer Failure, Explanation of Reasoning
[1] Auxiliary

Table 12: Prompt for functional role classification (continued).

You are given a question to an image and an answer paragraph to the question. You job is to label each of the sentence in the answer paragraph based on the type of information it provides as listed below. For each sentence, please assign all the information types that are applicable. Some sentences will not have any label assigned.

# Type: Image Content
# Definition: The sentence provides the visual information about the image content (e.g., setting, object, text, color, atmosphere, etc)
*This usually comes at the beginning of the sentence, to provide the overview of the image. This sentence often looks similar to typical image captions.
# Example: "The vehicle in the image appears to be a utility service truck, commonly used by contractors or maintenance departments."
# Example: "You are holding a pink bottle. Only some text on the bottle is visible but it is an Avon product. Some of the text on the bottle is in French."

# Type: Image Quality
# Definition: The sentence provides the information about the image quality.
# Example: "The image you've provided is quite blurry and the details are not clear, which makes it difficult to identify the jar's contents with certainty."

# Type: External Information
# Definition: The sentence offers general knowledge or facts that are not directly found from the image. It enriches the response with broader contextual or background information.
# Example: "The vehicle in the image appears to be a utility service truck, commonly used by contractors or maintenance departments."
# Example: "Expiration date is typically located near the top or side of the bottle or on the label itself."

Question: How many video games are in these shelves?
Answer:
[0] The image shows a shelf with a collection of video games.
[1] The exact number of games is not fully discernible due to the angle and focus of the photo, but I will do my best to estimate based on what is visible.
[2] The top shelf appears to have around 11 games visible, the middle shelf has approximately 30, and the bottom shelf has around 27.
[3] Please note that these numbers are rough estimates as the photo does not clearly show all games, and there may be additional games stacked or hidden behind the ones in front.
[4] Therefore, we can approximate that there are at least 68 video games visible in the shelves.
[5] However, the actual number could be slightly higher if there are multiple layers of games or games obscured from view.
Types:
[0] Image Content
[1] Image Quality
[2] Image Content
[3] Image Quality
[4] Image Content
[5] N/A

Question: What this?
Answer:
[0] The image you've provided is blurry, but it shows a part of a bottle with a blue label.
[1] There appears to be a barcode and possibly a brand logo, but due to the quality of the image, it's not possible to identify the contents of the bottle with certainty.
[2] If you have more information or a clearer picture, I could attempt to provide a more accurate identification.
Types:
[0] Image Content, Image Quality
[1] Image Content, Image Quality
[2] N/A

Table 13: Prompt for information source classification. Continued in Table 14.

Question: What is in this bottle?
Answer:
[0] The image is blurry, but it shows a bottle with a label that includes the word "MUSTARD".
[1] Given the color of the liquid visible through the bottle, it's likely to be a type of mustard sauce, possibly a honey mustard dressing or similar condiment based on its golden-brown hue.
[2] Mustard sauces are commonly used as a condiment or dressing for sandwiches, salads, and many other dishes.
Types:
[0] Image Content, Image Quality
[1] Image Content, Image Quality
[2] External Information

Question: What's in this?
Answer:
[0] This image is very close to the product and a little blurry.
[1] It shows goldfish crackers inside a clear ziploc container top and the bottom looks light blue holder.
[2] The container is on a light brown counter top.
Types:
[0] Image Quality
[1] Image Content
[2] Image Content

Question: Do this tie and shirt match?
Answer:
[0] No, the tie and shirt do not match.
[1] The shirt is a light green and white striped pattern, while the tie is a dark blue with light blue and white squares.
Types:
[0] Image Content
[1] Image Content

Question: What's this? Can you describe?
Answer:
[0] Shown in the photo is a tabletop with a Braille display/keyboard, several wires and charging chords, and a small bit of blue fabric, possible a bag or clothing.
[1] The make and model of the display is not visible but the keyboard is a matte Silver/grey.
[2] At the top are the braile input buttons, beneath which is the tactile readout.
[3] Jutting from the bottom, taking up the middle 1/3 of the display is another row of buttons, the center of which appears to be a directional pad.
Types:
[0] Image Content
[1] Image Content
[2] Image Content
[3] Image Content

Question: Hi, I'm going to start over. This is the place mat that are related to the napkins. First, I need to know if the stripes are vertical or horizontal.
Answer:
[0] The stripes on the place mat are horizontal, meaning they run from left to right across the width of the place mat.
Types:
[0] Image Content

Question: What is this?
Answer:
[0] The image shows a can of Goya brand red beans, specifically the "Frijoles Rojos" variety.
[1] This is a type of prepared beans that are commonly used in Latin American cuisine, both as a side dish and as an ingredient in many recipes.
[2] The can appears to be the typical size for such products, usually around 15 to 16 ounces, but the exact size is not visible in the image.
[3] Goya is a well-known brand that offers a variety of canned beans and other food products that are staples in many households.
Types:
[0] Image Content
[1] External Information
[2] Image Content, Image Quality, External Information
[3] External Information"""

Table 14: Prompt for information source classification (continued).

### A.2.3 Annotation Results

| Source (# of Annotations) | Confirmation | Answer | Ans. Failure | Auxiliary | Aux. Failure | Explanation | Suggestion | Misc. |
|---|---|---|---|---|---|---|---|---|
| **GPT-4V** (3393) | 11.29% | 22.58% | 10.43% | 17.27% | 0.53% | 24.23% | 11.97% | 1.71% |
| **Gemini** (1279) | 11.1% | 60.2% | 3.44% | 13.84% | 0.00% | 5.47% | 4.77% | 1.17% |
| **Expert** (2454) | 12.35% | 33.54% | 8.56% | 24.61% | 0.57% | 15.32% | 3.91% | 1.14% |
| **Total** (7126) | 11.62% | 33.10% | 8.53% | 19.18% | 0.45% | 17.79% | 7.9% | 1.42% |

Table 15: Distribution of functional roles in annotations

| Source (# of Annotations) | Image Content | Image Quality | External Information |
|---|---|---|---|
| **GPT-4V** (2472) | 49.8% | 23.67% | 26.54% |
| **Gemini** (1089) | 80.53% | 2.94% | 16.53% |
| **Expert** (2014) | 75.82% | 15.39% | 8.79% |
| **Total** (5575) | 65.20% | 16.63% | 18.17% |

Table 16: Distribution of information sources in annotations

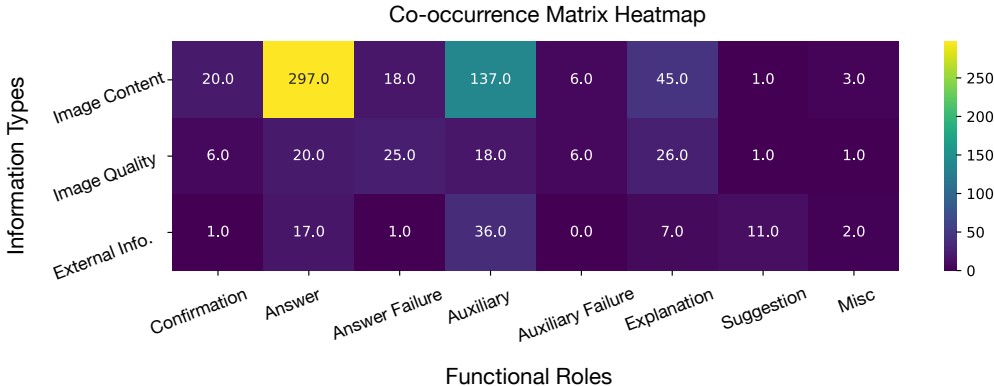

Figure 5: Co-occurrence heatmap displaying the association frequencies between functional roles and information sources. Darker colors indicate higher frequencies.

## A.3 Evaluation

### A.3.1 Automatic Evaluation

You are given a question, a single gold-standard reference answer written by expert, and a candidate answer. Please rate the accuracy of the candidate answer for the question considering the reference answer. Use a scale of 1-3, with 1 indicating an incorrect or irrelevant answer, 2 indicating an ambiguous or incomplete answer, and 3 indicating a correct answer. Give the rationale before rating. Follow the template of the examples following and always end the sentence with Rating: X.

Question: 'What is the color of the car?'
Reference answer: 'The color of the car is red.'
Candidate answer: 'red'
Output: The candidate answer is correct because both the reference answer and candidate answer mentions that the color is red.
Rating: 3

Question: 'What is the animal on the left?'
Reference answer: 'giraffe'
Candidate answer: 'giraffe'
Output: The candidate answer is correct because the reference answer and candidate answer are the same.
Rating: 3

Question: 'What's the weather like?'
Reference answer: 'rainy'
Candidate answer: 'The image displays a clear sky with a few small clouds, with the sun near the horizon suggesting it could be around sunrise or sunset. The sky has a subtle gradient, transitioning from a bright area near the sun to a darker blue further away. Due to the low exposure of the photo, the foreground including trees and a part of a building appear as silhouettes. The colors in the sky are muted, mostly displaying varying shades of blue without vibrant sunrise or sunset hues.'
Output: The candidate answer is incorrect because the weather is 'rainy' but the candidate answer does not mention it.
Rating: 1

Question: 'What is this picture about?'
Reference answer: 'The image shows a cartoon representation of two cats with a large pink heart in the background.'
Candidate answer: 'Two animated animal characters hugging each other.'
Output: The candidate answeris incomplete because it does not specify the type of animal and the background.
Rating: 2

Table 17: A full example of prompt used for LLM-based evaluation (Mañas et al., 2023). We adapted the few-shot example to account for diverse lengths of reference and candidate answers.

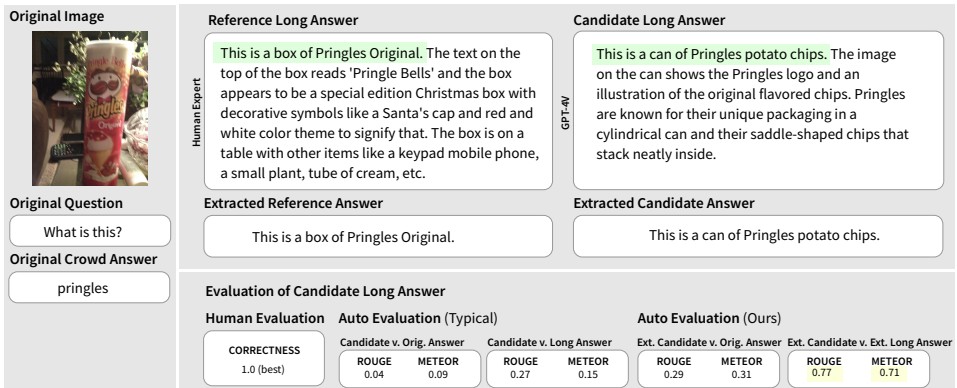

Figure 6: Example illustration of how functional roles can be utilized in automatic evaluation of long-form answers. While the human annotator marked the candidate long-form answer as correct, it shows low scores when directly compared against the short-form reference answer from VizWiz using traditional reference-based metrics (*e.g.*, ROUGE, METEOR). We show that extracting answer role sentences of long-form answers can mitigate this.

| ROUGE | | | | METEOR | | | | BERTScore | | | | GPT-4 (Mañas et al., 2023) | | | |
|---|---|---|---|---|---|---|---|---|---|---|---|---|---|---|---|
| $s_r$+$l_c$ | $s_r$+$l'_c$ | $l_r$+$l_c$ | $l_r$+$l'_c$ | $s_r$+$l_c$ | $s_r$+$l_c$ | $s_r$+$l'_c$ | $l_r$+$l_c$ | $l_r$+$l'_c$ | $s_r$+$l_c$ | $s_r$+$l_c$ | $s_r$+$l'_c$ | $l_r$+$l_c$ | $l_r$+$l'_c$ | $s_r$+$l_c$ | $s_r$+$l'_c$ |
| 0.15* | 0.22** | 0.05 | 0.16* | 0.18* | 0.26** | 0.16* | 0.17* | -0.17 | 0.11 | 0.06 | 0.16* | 0.34** | 0.35** | 0.46** | 0.43** |

Table 18: Pearson correlation between human judgment (Correctness) and popular automatic evaluation metric scores for 360 long-form answers($p < 0.05$ is marked with * and $p < 0.01$ is marked with **)

### A.3.2 Survey on BLV people's preferences in short vs long answers

Our work on long-form answers is motivated by the following reasons: (1) prior research in QA has shown that people have diverse preferences in length of answers (Choi et al., 2021), (2) many human studies with BLV people reveal that they want detailed descriptions (MacLeod et al., 2017; Salisbury et al., 2017; Gleason et al., 2020), (3) some questions in the VizWiz dataset even reveal this desire explicitly (*"Yes this is a Google Street View image, I need a detail of the description, as much as possible."*), and (4) BLV people are already active consumers of LFVQA services through applications like Be My AI (BeMyEyes, 2020).

| Metric | Ranking | | Relevance | | Helpfulness | | Plausibility | | Clarity | |
|---|---|---|---|---|---|---|---|---|---|---|
| | short | long | short | long | short | long | short | long | short | long |
| AVG | 0.25 | 0.75 | 2.18 | 2.6 | 1.93 | 2.65 | 2.38 | 2.68 | 1.92 | 2.7 |
| STDEV | 0.43 | 0.43 | 0.8 | 0.61 | 0.8 | 0.62 | 0.7 | 0.52 | 0.81 | 0.56 |

Table 19: Performance metrics with average and standard deviation for overall ranking and 4 ratings: relevance, helpfulness, plausibility, and clarity. Ranking ranges from 0-1 and ratings were collected with 3-point scale.

To compare BLV people's preferences in different lengths of answers to visual questions, we conducted a survey to understand how they evaluate short answers from crowd workers (from VizWiz) and long answers from expert describers (from VizWiz-LF) [12]. We recruited 8 BLV people who provided an overall ranking between short and long answers and 3-point scale ratings on *Relevance, Helpfulness, Plausibility,* and *Fleuncy*. Table 19 shows the results.

### A.3.3 BLV Participants

The experience of visual disabilities is individual and can affect people's preferences in image description Stangl et al. (2021); Huh et al. (2022). We report the onset and type of visual impairment, as these aspects may impact evaluations of long-form visual question answers (Table 20).

| PID | Gender | Age | Visual Impairment | Onset |
|---|---|---|---|---|
| 1 | 75 | Female | Totally blind | Congenital |
| 2 | 60 | Male | Totally blind | Acquired |
| 3 | 37 | Female | Legally blind | Congenital |
| 4 | 68 | Female | Totally blind | Congenital |
| 5 | 22 | Male | Totally blind | Acquired |
| 6 | 38 | Female | Totally blind | Congenital |
| 7 | 38 | Female | Totally blind | Congenital |
| 8 | 42 | Female | Totally blind | Acquired |
| 9 | 62 | Female | Legally blind | Congenital |
| 10 | 42 | Female | Totally blind | Congenital |
| 11 | 25 | Female | Totally blind | Congenital |
| 12 | 43 | Male | Totally blind | Congenital |
| 13 | 21 | Female | Totally blind | Congenital |
| 14 | 64 | Female | Totally blind | Congenital |
| 15 | 29 | Female | Totally blind | Congenital |
| 16 | 28 | Male | Totally blind | Congenital |
| 17 | 46 | Female | Legally blind | Congenital |
| 18 | 54 | Female | Totally blind | Congenital |
| 19 | 58 | Female | Totally blind | Congenital |
| 20 | 59 | Male | Legally blind | Acquired |

Table 20: BLV participants demographics

---

[12]Approved by our institution's Institutional Review Board (IRB). Participants were compensated $20 for their completion of the survey.

### A.3.4 Human Evaluation Setup

We conducted a 1.5 hour Zoom study with participants and shared the VQA data in Excel sheets. For BLV participants who could not access sheets with screen readers, researchers read out the questions and answers. We guided participants to evaluate long-form answers based on how well they will help users who do not have access to the image.

### A.4 Experiments on VQA Model Abstention

### A.4.1 Deriving Long-form Answer's Answerability Label

| Model | Abstained | Question/Answer |
|---|---|---|
| Expert | False | **Question:** What color is this? |
| | | **Answer:** *[Confirmation of Photo, Answer]* This fabric in this image is a light, minty green. |
| | True | **Question:** What is the title of this book? |
| | | **Answer:** *[Answer Failure]* The title of this book is not visible in this image. |
| GPT-4V | False | **Question:** Hi, I'm going to start over. This is the place mat that are related to the napkins. First, I need to know if the stripes are vertical or horizontal |
| | | **Answer:** *[Answer, Explanation of Reasoning]* The stripes on the place mat are horizontal, meaning they run from left to right across the width of the place mat. |
| | True | **Question:** What is the expiration date? |
| | | **Answer:** *[Answer Failure]* Sorry, I can't provide help with that request. |
| Gemini | False | **Question:** what color is this rubber band |
| | | **Answer:** *[Answer]* brown |
| | True | **Question:** any phone number written here. |
| | | **Answer:** *[Answer Failure]* The context does not mention anything about phone numbers. |

Table 21: Example question and long-form answer with their roles and derived abstention labels(§5). To determine whether a long-form answer abstained, we classify its sentence-level functional roles using the few-shot prompts and consider answers with *Answer Failure* roles as abstained and otherwise unabstained.

### A.4.2 Abstention Approaches

We provide example for each abstention approach in Table 22, 23, and 24. We also provide the configurations for answer generation in Table 25. For the vanilla, abstention, and self-correction approaches, we adhere to the default settings. For the Self-Consistency approach, we adjust the top-p and temperature settings to 0.7 to foster greater diversity in the generated outputs. This adjustment is based on our empirical testing, as the self-consistency approach requires aggregating multiple samples.

**Abstention Prompting**  Guiding instruction-tuned language models with principles can encourage models to provide trustworthy responses or choose not to respond when necessary (Zhou et al., 2023; Menick et al., 2022). We adopt similar strategies and prompt VQA models to determine whether a question is unanswerable (*e.g.*, overly blurry images) and abstain from answering in such cases (Full prompt in Table 22).

**Self-Correction**  Prior work has shown that LLMs and VLMs have the capability to identify and refine errors in their responses (Kadavath et al., 2022; Welleck et al., 2022; Cui et al., 2023; Olausson et al., 2023). Following Cui et al. (2023), we conduct three-stage prompting: we first input the image $i$ and the question $q$ without any prompt engineering to generate an initial response $a_0$; in a separate prompt, we provide the model with $q$, $i$ and $a_0$ and instruct it to review $a_0$ and output errors $e$; finally, we prompt the model with $q$, $i$, $a_0$ and identified errors $e$ and instruct it to generate a revised answer $a$ (full prompt in Table 23).

**Self-Consistency**  Self-consistency aggregates multiple sampled generations to provide an answer and demonstrated boosted performance for a variety of tasks (Wang et al., 2022; Chen et al., 2023d). We first generate 5 responses (using both vanilla and instruction prompts). Following (Chen et al., 2023d), we conduct aggregation with GPT-4 by prompting it with the question and the 5 responses, and instructing the model to output the most consistent response (full prompt in Table 24).

You are a helpful assistant that is answering questions about images for blind and low vision individuals. Do not hallucinate with incorrect answers if the question is unanswerable. A question is unanswerable if the provided image is too blurry, too bright or dark, obstructed, or ill-framed to correctly recognize. If a question asks about an object or asks to read a text but not visible in the image, it is also unanswerable. Abstain from answering if the question is unanswerable.
Question:
What's being displayed on the screen?

Table 22: A full example of prompt used in prompt instruction approach.

**Stage 1** (vanilla)
Question: What color is the jumper
Answer:
plaid

**Stage 2**
Review the question, image and your previous answer and find problems with your answer.
Question: What color is the jumper
Answer: plaid
Problems:

Plaid is a pattern, not a color.

**Stage 3**
Review the question, image and your previous answer and the problems with the answer. Based on the problems you found, improve your answer. Do not describe your previous answer or its mistake, but only output the revised correct answer.
Question: What color is the jumper
Answer: plaid
Problems: Plaid is a pattern, not a color.
Revised Answer:

plaid blue green white

Table 23: A full example of prompt used in self-correction approach.

I have generated the following responses to the question:
What is this box?
Response 0: Unanswerable. The image is too blurry to make out details of the box.
Response 1: The object is a roll of paper towels.
Response 2: The image is blurry and I cannot determine what the box contains.
Response 3: The image is blurry, and I cannot answer the question.
Response 4: The image is blurry, and I cannot determine what the box contains.
Evaluate these responses. Select the most consistent response based on majority consensus. Output only the following text "The most consistent response is Response X." (without quotes)

Table 24: A full example of prompt used in self-consistency approach.

| Model | Vanilla | Abstention | Self-Correction | Self-Consitecy |
|---|---|---|---|---|
| LLaVA | default (do_sample=False, temperature = 0) | default | default | do_sample=True, temperature=0.7 |
| BLIP-2 | default (do_sample=False, top-p=1, temperature =1) | default | default | do_sample=True, temperature=0.7, top-p = 0.7 |
| QWEN-VL-Chat | default (do_sample=True, top-p=0.3) | default | default | do_sample=True, top-p = 0.7 |
| InstructBLIP | default (do_sample=False, top-p=1, temperature =1) | default | default | do_sample=True, temperature=0.7 |

Table 25: VQA models' configurations in experiments of abstention techniques.

### A.4.3 Examples of Long-form Answers in *VizWiz-LF* dataset.

**VizWiz Unanswerable / Expert Answered**

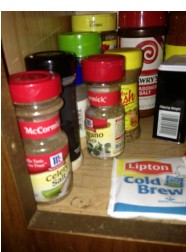

**Question**

*Can you please tell me all the spices that are in this cabinet?*

**VizWiz original**

*1. unanswerable*
*2. yes*
*3. unanswerable*
*4. lipton*
*5. celery salt oregano season salt mrs dash*
*6. celery salt oregano mrs dash seasoned salt*
*7. unanswerable*
*…*

**Expert**

*There are additional spices in this image that do not have a visable lable. The visible spices are as follows: round jar of Celery salt, round jar of oregano, round jar of Mrs. Dash seasoning, round and larger jar of Lawrys' seasoning salt, a rectangular jar of black pepper. There is also an individually packaged Lipton cold brew tea bag.*

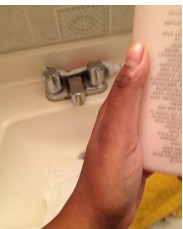

**Question**

*what is this?*

**VizWiz original**

*1. unsuitable*
*2. lotion*
*3. unsuitable*
*4. lotion*
*5. unsuitable*
*6. Bottle*
*7. hand holding product*
*…*

**Expert**

*You are holding a pink bottle. Only some text on the bottle is visible but it is an Avon product. Some of the text on the bottle is in French.*

**VizWiz Unanswerable / Expert Answered**

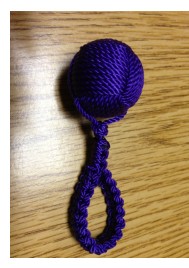

**Question**

*What is this and how can I find it?*

**VizWiz original**

*1. blue string ball on loop*
*2. dog toy*
*3. purple dog pull*
*4. cat toy*
*5. yarn cat toy*
*6. dog toy*
*7. this ball yarn you can find at craft shops*
*…*

**Expert**

*I am not quite sure what object you are showing. It looks like a trinket made of purple crocheting material with a ball on top that turns into a loop at the bottom.*

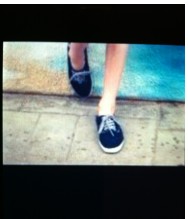

**Question**

*what is the brand of these shoes?*

**VizWiz original**

*1. they keds*
*2. vans*
*3. vans*
*4. high tops*
*5. keds*
*6. unanswerable*
*7. vans*
*…*

**Expert**

*I cannot tell the brand of the shoes. The image is blurry and the brand label is not on display.*

Figure 7: Examples of questions that VizWiz majority and expert describers did not agree on answerability.

### A.4.4  Results

| Model | Method | abs-acc | abs-prec | abs-recall | abs-f1 | abs-% |
|---|---|---|---|---|---|---|
| **Baseline** | Random | 0.50 | 0.34 | 0.34 | 0.34 | 38 |
| | Majority | 0.62 | 0.0 | 0.0 | 0.0 | 0 |
| | Expert | 0.73 | 0.68 | 0.57 | 0.62 | 32 |
| **GPT-4V** | vanilla | **0.77** | 0.66 | 0.82 | **0.73** | 47 |
| | instruction | 0.73 | 0.60 | **0.88** | 0.71 | **56** |
| | correction | 0.75 | **0.67** | 0.70 | 0.68 | 40 |
| | consistency | **0.77** | **0.67** | 0.77 | 0.72 | 43 |
| | instruct.+consist. | 0.72 | 0.59 | 0.87 | 0.70 | 43 |
| **Gemini** | vanilla | 0.66 | **0.78** | 0.14 | 0.24 | 7 |
| | instruction | 0.71 | 0.65 | 0.51 | 0.57 | 30 |
| | correction | 0.67 | 0.67 | 0.24 | 0.36 | 14 |
| | consistency | 0.65 | 0.76 | 0.13 | 0.21 | 6 |
| | instruct.+consist. | **0.72** | 0.66 | **0.54** | **0.60** | **32** |
| **LLaVA** | vanilla | 0.66 | 0.80 | 0.14 | 0.24 | 7 |
| | instruction | **0.78** | 0.70 | **0.74** | 0.72 | 41 |
| | correction | 0.67 | 0.69 | 0.23 | 0.34 | 12 |
| | consistency | 0.70 | **0.83** | 0.29 | 0.43 | 13 |
| | instruct.+consist. | 0.77 | 0.66 | **0.74** | **0.74** | **49** |
| **QWEN** | vanilla | 0.71 | **0.70** | 0.42 | 0.52 | 23 |
| | instruction | **0.76** | 0.64 | 0.84 | 0.72 | **50** |
| | correction | 0.65 | 0.53 | 0.62 | 0.57 | 45 |
| | consistency | 0.71 | **0.70** | 0.40 | 0.51 | 22 |
| | **instruct.+consist.** | 0.75 | 0.63 | **0.86** | **0.73** | 48 |
| **BLIP-2** | vanilla | 0.63 | 0.54 | 0.16 | 0.24 | 11 |
| | instruction | **0.70** | 0.59 | **0.73** | **0.65** | 47 |
| | correction | 0.63 | **0.65** | 0.10 | 0.16 | 6 |
| | consistency | 0.62 | 0.55 | 0.09 | 0.16 | 6 |
| | instruct.+consist. | 0.68 | 0.57 | **0.73** | 0.64 | **49** |
| **InstructBLIP** | vanilla | 0.63 | **0.63** | 0.10 | 0.17 | 6 |
| | instruction | 0.65 | 0.53 | **0.72** | 0.61 | **52** |
| | correction | 0.61 | 0.46 | 0.05 | 0.09 | 4 |
| | consistency | 0.62 | 0.52 | 0.05 | 0.09 | 4 |
| | instruct.+consist. | **0.69** | 0.58 | 0.66 | **0.62** | 44 |

Table 26: Performance of abstention strategies on six VLMs. Best results are in bold.

| Model | Approach | Question/Answer |
|-------|----------|-----------------|
| GPT-4V | Abstention prompting +consistency | **Question:** What kind of beer is this? |
| | | **Answer 1:** The image is blurry and it is difficult to read the full text on the can to provide the specific type of beer. However, the visible part of the label suggests it is a Grolsch beer due to the recognizable green can and the partial logo shown.
**Answer 2:** The beer in the image is Grolsch. You can tell by the distinctive green can and the Grolsch logo visible on it.
**Answer 4:** The beer in the image is Grolsch. This can be identified by the green color of the can and the distinctive Grolsch logo.
**Answer 5:** The beer shown is a Grolsch. It's recognizable by its green can and Grolsch logo.
**Final Answer:** The image is too blurry to accurately determine the specific type of beer.
**Answer 5:** The beer in the image is Grolsch. You can tell by the distinctive green can and the Grolsch logo visible on it. |
| Gemini | Abstention prompting +consistency | **Question:** What side will turn off the oven? Is it the left side or the right side to turn off oven?
**Answer 1:** The image is too blurry to tell which side will turn off the oven.
**Answer 2:** The side that will turn off the oven is the left side. The left side has a button with a power symbol.
**Answer 3:** The right side will turn off the oven.
**Answer 4:** The side that will turn off the oven is the left side. The left side has a button that says "OFF".
**Answer 5:** The right side will turn off the oven.
**Final Answer:** ADD HERE |
| Gemini | Self-consistency | **Question:** What is this?
**Answer 1:** It's a photo of a ceiling fan.
**Answer 2:** This is a photo of a black blanket.
**Answer 3:** It's a button.
**Answer 4:**
**Answer 5:**
**Final Answer:** It's a photo of a ceiling fan. |

Table 27: Examples of how VLMs reduce hallucinations in answers using abstention approaches.

