# OpenReview forum: "Long-Form Answers to Visual Questions from Blind and Low Vision People"
_colmweb.org/COLM/2024/Conference — COLM_

### Official Review · Reviewer_sGq5 · 2024-04-12

**Rating:** 5
**Confidence:** 4
**Ethics Flag:** 1

**Summary:**

VizWiz-LF presents a unique dataset of long-form answers from blind and low vision (BLV) users, shedding light on the discourse structure of such responses. It also highlights a discrepancy between model-generated and human-perceived correctness of answers to unanswerable questions. Additionally, the evaluation of VQA models provides insights for improving their performance and mitigating the risk of providing misleading answers.

**Reasons To Accept:**

- VizWiz-LF presents a unique dataset of long-form answers from BLV users, enriching the field of visual question answering.
- The evaluation of VQA models on their ability to abstain from answering unanswerable questions provides valuable insights for model development and deployment, aiming to mitigate the risk of providing misleading answers.

**Reasons To Reject:**

- poor presentation: This article is written with poor logic. For example, why we should understand how humans evaluate long-form answers, and there is no obvious relation between paragraph 5 and paragraph 6.
- The author needs to further clarify the difference between the BLV and traditional VQA dataset with long/short answers.
- Is any error in this sentence ‘’Vision language models can now generate long-form answers to questions about images – or, long-form visual question answers (LFVQA).’’
- What is the benefit of understanding the functional structure of long-form answers?
- Long-from answers from large models (such as GPT-4V) could introduce hallucinations and are not reliable.
- Human evaluation samples are limited.
- Human evaluation may not be reliable because people are subjective. Could authors provide the annotation detail (such as instructions for human evaluation)?
- More human evaluation samples should be provided .

---

> ### Author Rebuttal · Authors · 2024-05-31
>
> Dear Reviewer sGq5,
>
> Thank you for your thoughtful review!
>
> **Dataset contains answers from GPT-4V and Gemini**\
> See rebuttal to Reviewer LFVf with the same title.
>
> **Functional roles of LFVQA**\
> Prior research utilized functional roles in downstream NLP tasks including summarization [1], information retrieval [2], and generation of sentences of a specific functional role (e.g., explanation) [2]. To our knowledge, we are the first to design and annotate functional roles of answers to visual questions. We believe that understanding the functional roles of LFVQA can help us understand the behavior of a VQA model, compare its answer to that of other models or experts, and further utilize them for core answer extraction and fine-grained evaluation of long-form answers. We will clarify these motivations in Sections 1 and 3.3.
>
> **Human evaluation**\
> We conducted human evaluation to complement the limitations of the automatic evaluation. 6 automated metrics used to evaluate LFVQA have limitations (e.g., BERTscore can yield higher scores for semantically related but factually incorrect answers.) Human evaluation can bypass these deficiencies, provide flexibility (recognizing multiple valid answers), and assess diverse aspects including clarity or relevance. As human preferences vary, we conducted significance testing to ensure the trends we claim are reliable (Figure 2).
>
> We will also add full instructions provided to evaluators including evaluation scenario and definitions of the 4 evaluation criteria to A.1.2.
>
> **Connection to abstention experiment**\
> Our abstention experiment (paragraph 6) is motivated by the findings from the human evaluation (paragraph 5) which revealed that BLV users are more likely to be misled by hallucinations from VQA models. To better connect them, we updated the section orders and Introduction to describe automatic evaluation, human evaluation, and abstention experiments in order. We hope this better explains how human evaluation complements the automatic evaluation, and how the findings from the human evaluation motivate our abstention experiment.
>
> **Others**\
> We revised the sentence: “Vision language models can now generate long-form answers to questions about images, which we call long-form visual question answers (LFVQA).” We will further clarify the difference between BLV and the traditional VQA dataset in the paper.
>
> [1] “A Discourse-Aware …” Cohan et al., NAACL 2018.\
> [2] “Modeling Exemplification …” Wang et al., NAACL 2022.

---

> > ### Comment · Reviewer_sGq5 · 2024-06-04
> > **Thanks for your response**
> >
> > Some of my concerns are addressed, but other concerns still are not addressed (such as hallucinations). Therefore, I tend to improve my score mildly.

---

### Official Review · Reviewer_PViS · 2024-04-29

**Rating:** 8
**Confidence:** 4
**Ethics Flag:** 1

**Summary:**

The paper introduces VizWiz-LF, a new long-form visual question answering (VQA) dataset that focuses on visual questions posed by blind and low-vision (BLV) people. The datasets consists of 600 image-question pairs sampled from the VizWiz dataset. To ensure diversity of questions, the data is sampled from 4 question types, i.e., Identification, Description, Reading, and Others, which is based on the question taxonomy of VizWiz. Among the 600 image-question pairs, 230 are marked as unanswerable. For each image-question pair, 3 long-form answers are constructed -- one by human expert, one by GPT-4V, and the other by Gemini. The resulting answers from all annotations are more than 10x longer than the original VizWiz dataset. Each sentence in the long-form answers is annotated with functional roles (confirmation, answer, answer failure, auxiliary information, auxiliary information failure, explanation, suggestion, and miscelleneous) and information type (image content, image quality,  external information). To construct the annotation,  180 long-form answers are annotated by human through a three-way annotations achieving a high fleiss kappa score (0.76 for functional roles and 0.87 for information types). The rest of the data is annotated by GPT-4 through few-shot in-context prompt, which achieves F1-score of 0.74 compared to human F1-score of 0.79).

Using the generated data, the paper demonstrates that both human expert and GPT-4V produce diverse functional roles, while Gemini often onlty cover answer roles since the long-answer often only consists of a single sentence. Additionally, Gemini also rarely abstain to answer even when the question is unanswerable. Furthermore, the paper also shows that model-based approaches, i.e., Gemini and GPT-4V, contains a higher percentage of external information which does not appear on the image. Human evaluation is also conducted to measure the human preferences of the VizWizL-LF considering 4 factors including relevance. helpfulness, plausibility, and fluency. The human evaluation is conducted by 20 BLV and 20 sighted people. All evaluator groups favoured GPT-4V and human expert annotations over Gemini. While sighted people give low helpfulness score for Gemini due to incorrect information provided as an answer, BLV group rated high plausability for all annotations. An additional automatic evaluation using ROUGE, METEOR, and BERTScore is also conducted further showing the effectiveness of GPT-4V for generating long-form VQA. The paper further explore on abstention strategy which helps to the model to abstain from answering when the question is unanswerable.

**Questions To Authors:**

- Is there any human evaluation conducted to sighted and BLV people?
- The column gender and age should be swapped in Table 16 on Appendix A.3.1

**Reasons To Accept:**

- The paper introduces a simple method to extend the existing VQA dataset into a long-form VQA which is beneficial for blind and low-vision people
- The paper introduces a new long-form VQA dataset which is annotated with per sentence functional roles and information type, and showcases its applicability for analyzing the behaviour of synthetic data generated by LLMs.
- The paper provides an extensive analysis of the quality of the generated data through both human and automatic evaluations, and provides a clear in-depth analysis of the results
- The paper showcases an experiment on abstention to improve the ability of LLMs to abstain from answering unanswerable questions.
- The paper provides clear, easy-to-follow, and detailed enough writing (except for the result tables) on each section with complete detailed information in the appendix.

**Reasons To Reject:**

- The results tables are difficult to follow with too many numbers and have no clear highlight of which part readers should focus on
- Section 6 feels rather detached from the other contents of the paper
- It is not unclear how long-form VQA and VizWiz-LF can be beneficial for blind and low-vision people since the human evaluation is only conducted on long-form answers with no comparison on short-answer VQA.

---

> ### Author Rebuttal · Authors · 2024-05-31
>
> Dear Reviewer PViS,
>
> We appreciate your thoughtful review!
>
> **Presentation and clarity**\
> We will highlight the important numbers (e.g., bold highest values) in Table 4-5 to help readers focus on the key results. We will also swap the age and gender columns in Table 16.
>
> **Usefulness of LFVQA for BLV people**\
> We will clarify that our work on long-form answers is motivated by several reasons: (1) prior research in QA has shown that people have diverse preferences in length of answers [1], (2) many human studies with BLV people reveal that they want detailed descriptions [2, 3, 4], (3) some questions in the VizWiz dataset even reveal this desire explicitly (“Yes this is a Google Street View image, I need a detail, a detail of the description, as much as possible.”), and (4) BLV people are already active consumers of long-form VQA services through applications like Be My AI. Given the request for comparison, we additionally conducted a survey to understand how BLV people evaluate human short-form answers (from VizWiz) vs. human long-form answers (from VizWiz-LF) for overall preference and fine-grained criteria (relevance, helpfulness, plausibility, and clarity). BLV people preferred long-form answers to short-form answers (long: avg. $0.75$ vs short: avg. $0.25$, range: $0$-$1$, stdev. $0.43$) and this difference was significant (p-value: $3.66 \times 10^-8$). We will add a table of this survey results in the Appendix.
>
> **Abstention experiment section feels detached**\
> Our paper’s abstention experiment is motivated by the findings from the human evaluation study (Section 4) that BLV users are more likely to be misled by false information hallucinated by VQA models, and that unanswerable questions are prevalent in VQA datasets for BLV users. To better connect Section 6 to other parts of the paper, we will revise the paper to change the section orders: 4. Automatic Evaluation, 5. Human Evaluation, and 6. Experiments on VQA Model Abstention. We hope this better explains how we conducted human evaluation studies to complement the automatic reference-based evaluation, and how the findings from the human evaluation studies motivate our experiments on VQA model abstention.
>
> [1] “Decontextualization…” Choi et al., TACL 2021\
> [2] “Understanding Blind …” MacLeod et al., CHI 2017\
> [3] “Toward Scalable …” Salisbury et al., HCOMP 2017\
> [4] “Twitter A11y …” Gleason et al., CHI 2020

---

> > ### Comment · Reviewer_PViS · 2024-06-04
> >
> > Thank you for addressing all the questions and comments. Upon considering all the reviews and authors' responses, I have decided to keep my current score.

---

### Official Review · Reviewer_1BBk · 2024-05-25

**Rating:** 7
**Confidence:** 4
**Ethics Flag:** 1

**Summary:**

This paper presents VizWiz-LF, a VQA dataset consisting of long-form answers to visual questions posed by blind and low vision (BLV) users. VizWiz-LF contains 1.8k long-form answers with a total of 5.2k sentences collected from human expert describers and two vision language models (GPT-4V and Gemini). The authors show that long-form answers contain information beyond the answer such as explanations and suggestions, and also conduct a human evaluation of the long-form answers with BLV and sighted people. Overall, the paper presents a useful benchmark, and has insightful findings.

[Update: Thank you for the clarification! My concerns have been sufficiently addressed by the author rebuttal, and I'm increasing my score.]

**Questions To Authors:**

* Are the GPT-4V and Gemini answers intended to be used to evaluate other models on this dataset? If so, why not just compare model outputs to human ground-truth responses?
* Is there a reason why the six open-source models were not evaluated on the long-form generative VQA task?

**Reasons To Accept:**

* This paper tackles an important problem of evaluating long-form VQA, which is underlooked in the literature.
* The paper is well-presented and clearly written.
* The analysis of unanswerability and VQA model abstention, in particular, is novel and important.
* The findings that a human expert and GPT-4V’s answers contain different functional roles is interesting, and I thought that investigating the communicative goal that each sentence plays within a long-form answer is a much-needed perspective on VQA.

**Reasons To Reject:**

* It isn’t clear to me why answers in the dataset were elicited from GPT-4V and Gemini as well as human experts? Using GPT-4V and Gemini answers as ground-truths could potentially introduce some biases and could lead to quality issues in the dataset.
* Related to this concern, is there only one human-elicited long form answer per visual question? My concern is that this will make it difficult to evaluate other models with only one ground-truth for comparison.
* 600 questions within the dataset is on the smaller side, although I believe this shortcoming is made up by the detailed, in-depth qualitative analysis and human studies.
* Only two closed-source models (Gemini and GPT-4V) were evaluated on the dataset, and the six open-source models were evaluated only on the task of abstention, and not on generating long-form answers.

---

> ### Author Rebuttal · Authors · 2024-05-31
>
> Dear Reviewer 1BBk,
>
> Thank you for your helpful review!
>
> **Dataset contains answers from GPT-4V and Gemini**\
> See rebuttal to Reviewer LFVf with the same title.
>
> **Dataset collects only one expert answer per question**\
> We prioritized our annotation budget on collecting one expert long-form answer per visual question to create a larger dataset (600 questions) over collecting multiple long-form answers for a smaller dataset. This choice aligns with numerous other popular dataset publications [2, 3] that only collect one annotation per example because they rely on expert annotators. In our case, people with experience describing images for BLV people were challenging to recruit and expensive to hire (see A.1.2). Many dataset papers that collect redundant annotations do so because they rely on less expensive, less-trained, anonymous crowd-workers, or contain noisy data crawled from online resources [4]. Still, we agree that collecting more expert annotations could be valuable. Similar to prior work [1], future work can augment our VizWiz-LF dataset with more ground truth references and perform human correlation studies. We will mention this in Section 2.
>
> **Dataset size**\
> While our dataset has a smaller number of questions due to cost and API rate limits, the dataset still represents a variety of question types as we intentionally evenly sampled questions for each of four question types (Identification, Description, Reading, Others). We collected multi-sentence answers for each of the questions from multiple sources and provided a thorough analysis of our dataset with detailed qualitative analysis and human studies.
>
> **Open-source models not evaluated on long-form answers generation**\
> We additionally conducted automatic evaluation of six open-source models using multiple reference-based metrics (ROUGE, METEOR, BERTScore, LAVE) and will add these results to A.3.2. We chose to perform human evaluation only on human generated long-form answers and two models GPT-4V and Gemini (demonstrated as state-of-the-art in image understanding tasks [5, 6]), due to the cost of human evaluation and the difficulty of recruiting BLV participants.
>
> [1] “Investigating Evaluation …” Gupta et al., SIGDIAL 2019\
> [2] “Vizwiz-priv …” Gurari et al., CVPR 2019\
> [3] “Scene Parsing …” Zhou et al., CVPR 2017\
> [4] “ELI5 …” Fan et al., ACL 2019\
> [5] “MMMU: A Massive…” Yue et al., CVPR 2024\
> [6] “Gemini: A Family of Highly Capable Multimodal Models, Team et al.

---

> > ### Comment · Reviewer_1BBk · 2024-06-04
> > **Reply to Author R**
> >
> > Thank you for the clarification! My concerns have been addressed, and I'm increasing my score.

---

### Official Review · Reviewer_LFVf · 2024-05-30

**Rating:** 6
**Confidence:** 3
**Ethics Flag:** 1

**Summary:**

The paper presents a dataset, called VizWiz-LF, which is an extension of the VizWiz dataset with long form answers. VizWiz-LF consists of 1.8k long-form answers with a total of 5.2k sentences collected from human expert describers and two vision language models (GPT-4V and Gemini). The analysis suggested that long form answers contain explanations and suggestions which could be useful for BLV and sighted people. The paper evaluated six VQA models to check the ability of abstaining from answering unanswerable questions.

**Questions To Authors:**

The reliability of the dataset largely depends on the accuracy of the task performance of the crowd workers. I may have missed but does the paper provide details about the crowd workers? What are their background, how they are compensated, and how much time they took to finish their tasks?

**Reasons To Accept:**

- The paper is on a very important topic. The paper is well motivated and the organization of the paper is pretty good.
- Experiments are quite comprehensive. The analysis and human evaluation are very helpful. It will help future works.

**Reasons To Reject:**

This is a straight-forward dataset paper that extends a previously proposed dataset. Some of the annotations are model based, therefore, there may be some noise/inaccuracies, however, the human evaluation shows such chances are quite low. Overall, this is a good work with a very limited novelty.

---

> ### Author Rebuttal · Authors · 2024-05-31
>
> Dear Reviewer LFVf,\
> \
> Thank you for your thoughtful review! We are encouraged that the review recognized the importance of our problem and our comprehensive evaluations and experiments.
>
> **Dataset contains answers from GPT-4V and Gemini**\
> We included model generated answers as well as expert answers in the dataset so that it is easy to compare the structure (functional roles) and quality (human ratings) of the long-form answers from multiple sources. We agree that GPT-4V and Gemini can introduce noise/inaccuracies. Thus, we collected the model answers to conduct an in-depth analysis in this new domain of long-form visual question answering rather than using the model answers as ground truth to evaluate other models. We will clarify the goal of our dataset in the introduction and conclusion. While we already discuss the limitations of potential hallucinations in the dataset in our ethics statement, we will also mention these limitations when we introduce the dataset in Section 3.
>
> **Expert describers’ information**\
> We recruited expert describers with experiences in describing images for BLV people using Upwork (e.g., professional audio description writers, full-time visual interpreters). We compensated describers with their hourly rate (28 USD, STDEV = 8), and describers answered 30 questions each, spending an average of 3.9 minutes (STDEV = 1.2 minutes) per visual question. We included compensation and task time in Appendix A.1.2, and we will add a reference to this A.1.2 when we introduce the description task in Section 3.2.

---

> > ### Comment · Reviewer_LFVf · 2024-06-04
> > **Thank you for the rebuttal**
> >
> > Thank you for addressing the comments. After careful consideration, I have decided to retain my score.

---

### Decision · Program_Chairs · 2024-07-10

**Decision:**

Accept

**Comment:**

This paper introduces VizWiz-LF, an extension of VizWiz with 600 questions and long answers.  VizWiz-LF (like its predecessor) was created with the goal of helping blind or low vision users learn what is in images. This is an important area for generative AI. The dataset seems to have been thoughtfully created and motivated. Some concerns were raised during the reviewing process about the use of model-generated outputs in parts of the dataset, but the authors addressed these concerns to the satisfaction of at least most of the reviewers. The paper is also well-written, and it includes a substantial number of experimental evaluations.